# Reactive oxygen species-dependent Toll/NF-κB activation in the *Drosophila* hematopoietic niche confers resistance to wasp parasitism

Isabelle Louradour[†], Anurag Sharma[‡], Ismael Morin-Poulard[§], Manon Letourneau, Alain Vincent, Michèle Crozatier*, Nathalie Vanzo*

Centre de Biologie du Développement, Centre de Biologie Intégrative, Université de Toulouse, CNRS, UPS, Toulouse, France

**Abstract** Hematopoietic stem/progenitor cells in the adult mammalian bone marrow ensure blood cell renewal. Their cellular microenvironment, called 'niche', regulates hematopoiesis both under homeostatic and immune stress conditions. In the *Drosophila* hematopoietic organ, the lymph gland, the posterior signaling center (PSC) acts as a niche to regulate the hematopoietic response to immune stress such as wasp parasitism. This response relies on the differentiation of lamellocytes, a cryptic cell type, dedicated to pathogen encapsulation and killing. Here, we establish that Toll/NF-κB pathway activation in the PSC in response to wasp parasitism non-cell autonomously induces the lymph gland immune response. Our data further establish a regulatory network where co-activation of Toll/NF-κB and EGFR signaling by ROS levels in the PSC/niche controls lymph gland hematopoiesis under parasitism. Whether a similar regulatory network operates in mammals to control emergency hematopoiesis is an open question.
DOI: https://doi.org/10.7554/eLife.25496.001

**\*For correspondence:**
michele.crozatier-borde@univ-tlse3.fr (MC);
nathalie.vanzo@univ-tlse3.fr (NV)

**Present address:** [†]National Institutes of Health, Bethesda, United States; [‡]Department of Biomedical Sciences, NU Centre for Science Education and Research, Nitte University, Mangalore, India; [§]Hubrecht Institute, Utrecht, Netherlands

**Competing interests:** The authors declare that no competing interests exist.

## Introduction

In adult mammals, Hematopoietic Stem and Progenitor Cells (HSPCs) present in the bone marrow are at the origin of blood cell production throughout an individual's life. Self-renewal, proliferation and differentiation of HSPCs are under the control of the bone marrow microenvironment called the 'niche' which has emerged as a key regulator of hematopoiesis (*Calvi and Link, 2015*; *Morrison and Scadden, 2014*). While considerable effort has been directed toward uncovering the mechanisms that regulate HSPC/niche communication under normal conditions, processes that take place in response to infection and inflammation remain more elusive (*Kobayashi et al., 2016*; *Zhao and Baltimore, 2015*). With the discovery that many transcriptional regulators and signaling pathways controlling blood cell development are conserved between insects and humans, *Drosophila* has become an important model to investigate hematopoiesis under normal conditions and in response to immune stress (*Letourneau et al., 2016*; *Martinez-Agosto et al., 2007*).

Immune cells in *Drosophila*, called hemocytes, contribute to innate immune defense (*Lemaitre and Hoffmann, 2007*). Two phases of hematopoiesis occur in *Drosophila* (*Gold and Brückner, 2015*; *Holz et al., 2003*). A first wave of immune cell production takes place during embryogenesis and generates two types of hemocytes, plasmatocytes and crystal cells, which are involved in phagocytosis and melanisation/wound healing, respectively (*Lemaitre and Hoffmann, 2007*). Both cell types increase in numbers at larval stages and either circulate in the hemolymph or aggregate into pockets under the epidermis (sessile hemocytes) (*Makhijani et al., 2011*; *Márkus et al., 2009*). A second wave of de novo hematopoiesis takes place in a larval specialized

hematopoietic organ, the lymph gland, a multi-lobed organ which develops along the anterior cardiac tube, the aorta region (*Jung et al., 2005*; *Krzemien et al., 2010a*; *Lanot et al., 2001*). During the third larval stage, the anterior lobes of the mature lymph gland are composed of a medullary zone (MZ) containing progenitors (prohemocytes), a cortical zone (CZ) composed of differentiating plasmatocytes and crystal cells, and the posterior signaling center (PSC) (*Jung et al., 2005*). During metamorphosis, lymph gland rupture releases all hemocytes into circulation (*Grigorian et al., 2011*; *Lanot et al., 2001*). A third blood cell type, called lamellocyte, differentiates in emergency situations, such as wasp parasitism, from lymph gland progenitors and circulating/sessile hemocytes (*Anderl et al., 2016*; *Crozatier et al., 2004*; *Honti et al., 2010*; *Lanot et al., 2001*; *Márkus et al., 2009*; *Stofanko et al., 2010*). Lamellocytes are specialized hemocytes which mediate the encapsulation and killing of pathogens too large to be phagocytosed. Following wasp egg laying inside the *Drosophila* larva, the egg is recognized as a foreign body. This leads to differentiation of lamellocytes at the expense of other immune cells (*Honti et al., 2010*; *Krzemień et al., 2007*; *Márkus et al., 2009*; *Stofanko et al., 2010*), the premature dispersal of the lymph gland and the release of lamellocytes into the hemolymph (*Crozatier et al., 2004*; *Sorrentino et al., 2002*).

The PSC plays a major role to control the immune response to wasp parasitism (*Benmimoun et al., 2015*; *Crozatier et al., 2004*; *Krzemień et al., 2007*; *Oyallon et al., 2016*; *Sinenko et al., 2011*). In the absence of PSC, such as in *collier* (*col/kn/ebf*) mutants, or upon its genetic ablation, massive differentiation of lamellocytes fails to occur. It was recently established that wasp parasitism increases Reactive Oxygen Species (ROS) levels in PSC cells, leading to the secretion of Spitz (Spi), one ligand of the Epidermal Growth Factor Receptor (EGFR) signaling pathway (*Sinenko et al., 2011*). These authors proposed that Spi released from PSC cells activates the EGFR pathway in circulating hemocytes, inducing them to differentiate into lamellocytes.

Several independent studies established that the evolutionarily conserved Toll/NF-κB pathway regulates *Drosophila* hematopoiesis both under homeostatic and wasp parasitism conditions (*Gao et al., 2016*; *Gueguen et al., 2013*; *Matova and Anderson, 2006*, *2010*; *Qiu et al., 1998*; *Schmid et al., 2014*; *Sorrentino et al., 2004*). However, how Toll/NF-κB controls wasp egg encapsulation remains unknown. Here, we show that this pathway is activated in PSC cells in response to egg laying by *Leptopilina boulardi* wasps and non-cell autonomously controls lamellocyte differentiation in the lymph gland. This, in turn, leads to lymph gland disruption and *in fine* to successful wasp egg encapsulation. We establish that a regulatory network linking ROS to Toll/NF-κB in the PSC and to EGFR/Erk signaling in lymph gland progenitors controls the lymph gland response to parasitism and the success of wasp egg neutralization.

## Results

### Toll/NF-κB signaling controls the lymph gland response to wasp infection

In *Drosophila*, Toll/NF-κB pathway activation by the ligand Spätzle (Spz) leads to the recruitment of an adaptor complex that consists of three death domain-containing proteins, Myd88, Tube, and Pelle (Pll). This complex triggers phosphorylation and degradation of the IκB factor Cactus (Cact) and ultimately the release of the NF-κB transcription factors Dorsal (Dl) and Dorsal-related immunity factor (Dif) that translocate into the nucleus where they activate target gene transcription (*Lemaitre and Hoffmann, 2007*). When parasitized by the wasp *Leptopilina boulardi*, mutant larvae for either *Toll* (*Tl*), *tube* or *pll* fail to encapsulate wasp eggs (*Sorrentino et al., 2004*). To determine whether this failure is linked to impaired lymph gland response to parasitism, we analyzed lymph gland disruption in mutants of the Toll/NF-κB pathway. 30 hr post-wasp egg laying, 93% of wild-type (*wt*) larvae lymph glands are disrupted, that is, lymph gland cells are released into circulation (*Figure 1A,C*). In contrast, in *pll²/pll⁷* mutants, 98% of the lymph glands remain intact (*Figure 1B,C*). Likewise, more than half of lymph glands are intact in parasitized *Dif¹* mutant, while the others start disrupting, that is, only some cells at the periphery of the anterior lobes disperse (*Figure 1D,F*). At a later time point, 48 hr post-parasitism, *Dif* and *pll* mutant lymph glands have disrupted (*Figure 1H, I*), similar to previously reported for *Tl* and *tube* mutants (*Sorrentino et al., 2004*). *dl¹* lymph glands, however, disrupt as in control larvae (*Figure 1E,G*). Corroborating these findings, lymph gland dispersal is delayed 30 hr post-parasitism in *Dif¹/Df(2L)J4* larvae - *Df(2L)J4* removes both *Dl* and *Dif* -

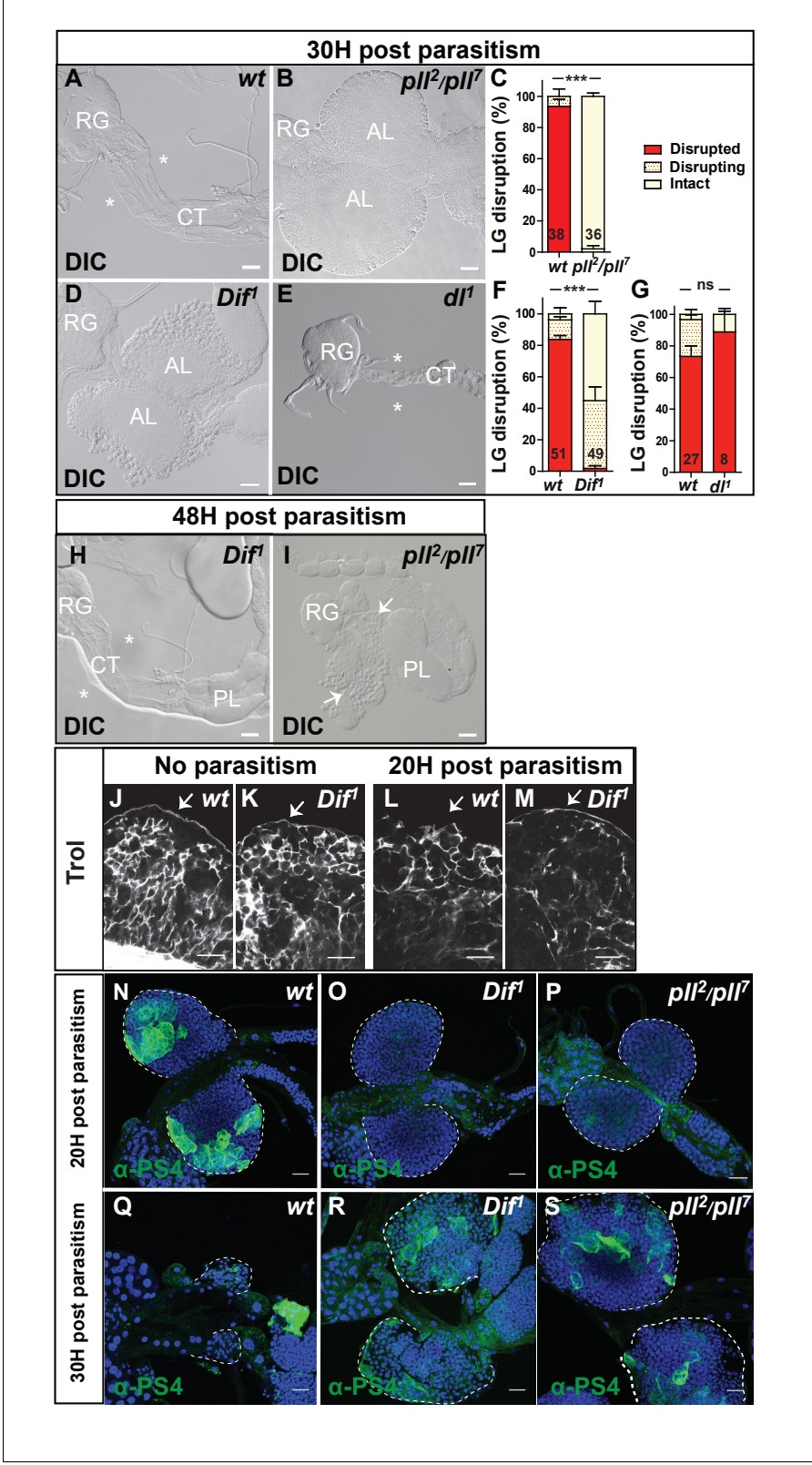

**Figure 1.** Dif-dependent Toll/NF-κB signaling is required for correct timing of lymph gland dispersal in response to wasp parasitism. (**A, B, D, E**) Differential interference contrast (DIC) microscopy of lymph gland anterior lobes 30 hr post-parasitism. Lymph glands were classified into three groups (see Materials and methods): Disrupted (*) in *wt* (wild type, (**A**) or *dl*[1] mutant (**E**); intact in *pll*[2]/*pll*[7] mutant (**B**); disrupting in *Dif*[1] mutant (**D**). Representative lymph

*Figure 1 continued on next page*

*Figure 1 continued*

glands are shown. Ring gland, RG; cardiac tube, CT; anterior lobe, AL; posterior lobe, PL. (C, F, G) Quantification of lymph gland disruption (%). The error bars correspond to SEM, ***p<0.001 and ns (not significant) (Pearson's Chi-squared test). In this and subsequent similar analyses, numbers of lymph glands analyzed are indicated on histograms. (H, I) DIC microscopy of lymph glands in *Dif$^1$* (H) and *pll$^2$/pll$^7$* (I) mutants 48 hr post-parasitism. Disrupted (*) and disrupting (arrows) lymph glands are shown. (J–M) Perlecan/Trol (white) immunostaining in control (J, L) and *Dif$^1$* anterior lobes (K, M). (N–S) αPS4 (green) immunostaining labels lamellocytes in lymph glands in *wt* (N, Q), *Dif$^1$* (O, R) and *pll$^2$/pll$^7$* (P, S) mutants. 20 hr post-parasitism, more than 65% (n = 24) of *wt* lymph glands displayed lamellocytes, whereas none (n = 24) or less than 15% (n = 18) contained lamellocytes in *Dif$^1$* and *pll$^2$/pll$^7$* mutants, respectively. 30 hr post-parasitism, *Dif$^1$* and *pll$^2$/pll$^7$* mutant lymph glands have lamellocytes, whereas most *wt* lymph glands have disrupted. In this and subsequent figures (unless specified), the scale bar represents 20 µm and nuclei are labeled with TOPRO3 or Draq5.

DOI: https://doi.org/10.7554/eLife.25496.002

The following source data and figure supplements are available for figure 1:

**Source data 1.** Lymph gland disruption quantification.
DOI: https://doi.org/10.7554/eLife.25496.005
**Figure supplement 1.** Dif but not Dorsal is required for proper lymph gland disruption in response to wasp parasitism.
DOI: https://doi.org/10.7554/eLife.25496.003
**Figure supplement 1—source data 1.** Lymph gland disruption quantification.
DOI: https://doi.org/10.7554/eLife.25496.004

compared to +/*Df(2L)J4* controls and *dl$^1$*/*Df(2L)J4* larvae (*Figure 1—figure supplement 1*). Delayed lymph gland disruption is also observed in *Dif$^1$* but not *dl$^1$* isogenic mutant larvae, further ruling out a genetic background effect (*Figure 1—figure supplement 1*). Collectively, these data show that Toll/NF-κB signaling is required for the proper timing of lymph gland dispersal in response to wasp infection and that Dif is the only NF-κB factor involved.

Lymph gland dispersal in response to wasp parasitism is associated with the loss of the basement membrane surrounding the organ, allowing the release of lamellocytes into circulation (*Sorrentino et al., 2002*). In contrast to control larvae (*Figure 1J,L*), immunostaining of parasitized *Dif$^1$* mutant lymph glands for an extracellular matrix (ECM) protein, the heparan sulfate proteoglycan Perlecan/Trol (Terribly Reduced Optic Lobes) (*Dragojlovic-Munther and Martinez-Agosto, 2013*; *Grigorian et al., 2013*) indicates the presence of a continuous ECM layer surrounding the anterior lobes 20 hr post parasitism (*Figure 1K,M*). Thus, the delayed lymph gland dispersal observed in *Dif$^1$* mutant is associated with basement membrane persistence. We then analyzed lamellocyte differentiation using α-PS4 immunostaining (*Krzemień et al., 2007*). 20 hr post-parasitism, lamellocytes are detected in control larvae (*Figure 1N*), whereas none are found in *Dif$^1$* and *pll$^2$/pll$^7$* mutants (*Figure 1O,P*). 30 hr post-parasitism, while most control lymph glands have already released lamellocytes into circulation (*Figure 1Q*), few lamellocytes are detected in mutant lymph glands (*Figure 1R–S*). Altogether, these data show that Toll/NF-κB signaling, while not required for lamellocyte differentiation per se, is essential for the timely differentiation of lamellocytes in the lymph gland, breakdown of its basement membrane, and lamellocyte release into the hemolymph.

## Lymph gland lamellocytes are required for successful wasp egg encapsulation

Upon parasitism, lamellocytes mediate wasp egg encapsulation, which is critical for host survival since it prevents hatching of wasp larvae (*Crozatier et al., 2004*). Wasp egg hatching, which reliably reflects the success of the immune response (*Mortimer et al., 2013*; *Schmid et al., 2014*; *Vanha-Aho et al., 2015*), is impaired in either *Dif$^1$* (*Figure 2A–C*, *Figure 2—figure supplement 1*) or *pll$^2$/pll$^{21}$* and *Tl$^{rv1}$/Tl$^r$* mutants (*Figure 2—figure supplement 1*) (*Sorrentino et al., 2004*) but not in *dl$^1$* mutants (*Figure 2—figure supplement 1*), indicating that Dif-mediated activation of Toll/NF-κB pathway is required for wasp egg encapsulation. These data, together with the delay in lymph gland disruption observed in *Dif* and *pll* mutants (*Figure 1*), strongly suggest that lymph gland-derived lamellocytes are essential for successful wasp egg encapsulation. To strengthen this conclusion, we measured the percentage of circulating lamellocytes just prior, and immediately following lymph

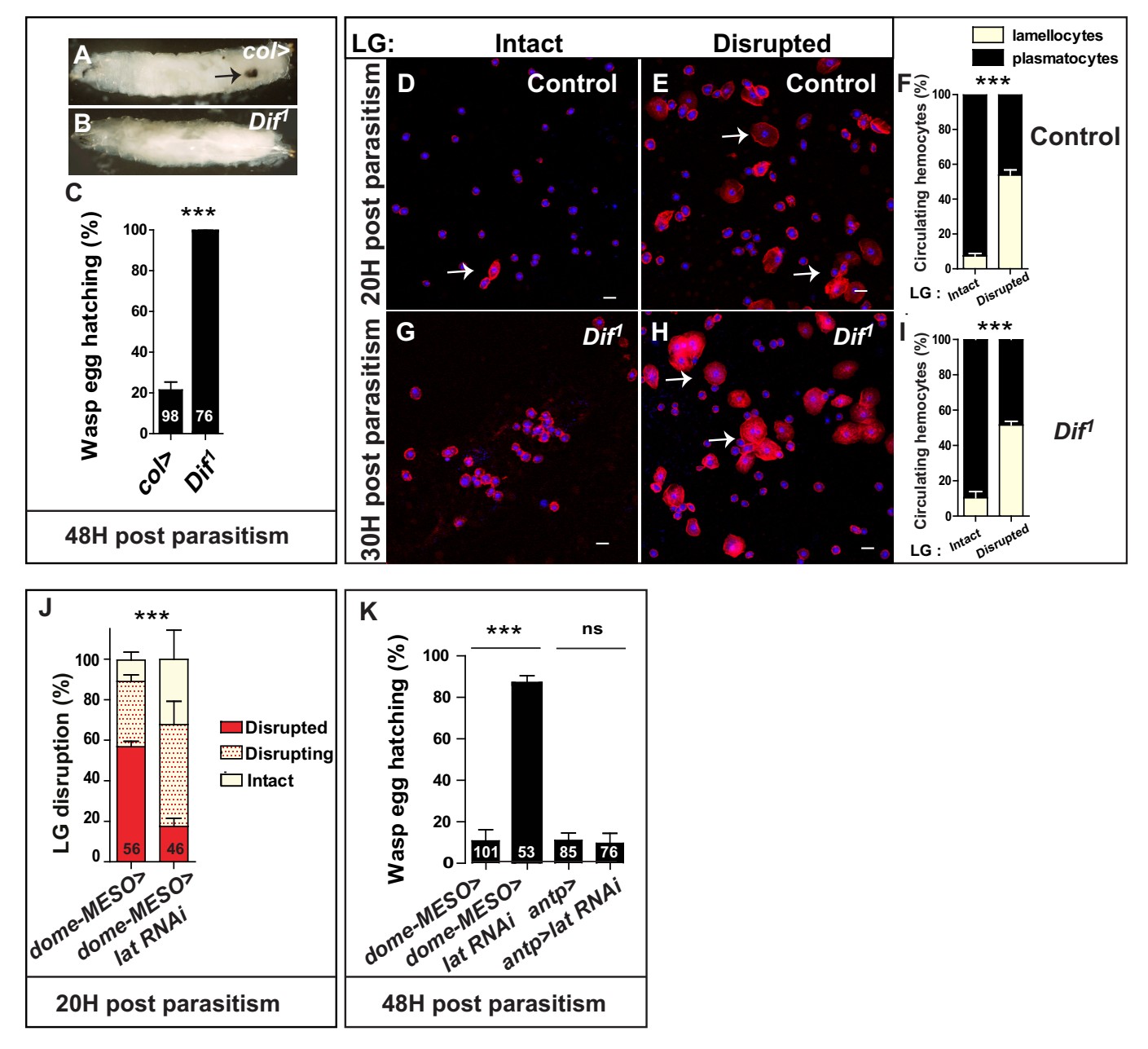

**Figure 2.** Dif, lymph gland lamellocyte differentiation and lymph gland disruption are required for wasp egg neutralization. (A, B) Brightfield images of control (*col>*, A) and *Dif¹* mutant larvae 48 hr post parasitism (B). The arrow indicates a melanotic capsule. (C) Quantification (%) of wasp egg hatching in control (*col>*) and *Dif¹* mutant. In this and subsequent similar analyses, numbers of (infected) larvae analyzed are indicated on histograms. (D–I) Phalloidin (red) staining of circulating hemocytes in control (*hml>*, D, E) and *Dif¹* mutant (G, H) just prior (D, G) and following (E, H) lymph gland disruption. Arrows indicate discoidal-shaped lamellocytes that are larger than plasmatocytes. (F, I) Quantification (%) of circulating plasmatocytes and lamellocytes in control (F) and *Dif¹* mutant (I). Blue, TOPRO3. (J) Quantification of lymph gland disruption (%) in control larvae (*dome-MESO>*) and when *latran* is down-regulated in lymph gland progenitors (*dome-MESO > lat RNAi*). (K) Quantification of wasp egg hatching in control larvae (*dome-MESO>*) or (*antp>*) and when *latran* is down-regulated in lymph gland progenitors (*dome-MESO > lat RNAi*) or in PSC cells (*antp > lat RNAi*). Error bars correspond to SEM, ***p<0.001 and ns (not significant) (Pearson's Chi-squared test).

DOI: https://doi.org/10.7554/eLife.25496.006

The following source data and figure supplements are available for figure 2:

**Source data 1.** Wasp egg hatching quantification.

DOI: https://doi.org/10.7554/eLife.25496.010

**Source data 2.** Circulating hemocyte quantification.

*Figure 2 continued on next page*

*Figure 2 continued*

DOI: https://doi.org/10.7554/eLife.25496.011

**Source data 3.** Lymph gland disruption quantification.

DOI: https://doi.org/10.7554/eLife.25496.012

**Source data 4.** Wasp egg hatching quantification.

DOI: https://doi.org/10.7554/eLife.25496.013

**Figure supplement 1.** Toll/NF-κB signaling is required for wasp egg encapsulation.

DOI: https://doi.org/10.7554/eLife.25496.007

**Figure supplement 1—source data 1.** Wasp egg hatching quantification.

DOI: https://doi.org/10.7554/eLife.25496.008

**Figure supplement 2.** At 18°C, lymph gland disruption and the presence of lamellocytes in circulation are delayed.

DOI: https://doi.org/10.7554/eLife.25496.009

gland disruption. Before disruption, which occurs around 20 hr post-parasitism in control larvae and 30 hr post parasitism in *Dif[1]* mutants, less than 10% of circulating hemocytes are lamellocytes (*Figure 2D,F,G,I*). This indicates that a few lamellocytes differentiate from sessile/circulating hemocytes prior to lymph gland dispersion. Following lymph gland disruption, however, more than 50% of circulating hemocytes are lamellocytes in both control and *Dif[1]* mutant larvae (*Figure 2E,F,H,I*). We thus conclude that lymph gland disruption causes a rapid and major increase in the number of circulating lamellocytes and that this is delayed in Toll/NF-κB mutants. A delay in both lymph gland disruption (data not shown) and in the increase numbers of circulating lamellocytes, is observed in parasitized wild-type larvae raised at 18°C (*Figure 2—figure supplement 2*), indicating that the lymph gland response is temperature-dependent.

To further investigate the contribution of lymph gland-derived lamellocytes to the success of wasp egg encapsulation, we specifically impaired lymph gland progenitor differentiation and analyzed the consequence on wasp egg encapsulation. We had previously established that the dominant-negative JAK/STAT receptor *latran (lat)* (*Eye transformer* [*Kallio et al., 2010*]), down-regulates JAK/STAT signaling in lymph gland progenitors following wasp attack, allowing massive differentiation of lymph gland lamellocytes (*Makki et al., 2010*). Accordingly, we found that knocking-down *lat* specifically in lymph gland progenitors using double-stranded RNA-mediated interference (RNAi) (*dome-MESO > lat* RNAi) impairs lymph gland dispersal post-parasitism (*Figure 2J*). Moreover, 90% of wasp eggs hatched when *lat* was inactivated in progenitor cells (*dome-MESO > lat* RNAi) compared to 10% in controls (*dome-MESO>*), whereas no defect was observed when *lat* was inactivated in PSC cells (*antp > lat* RNAi) (*Figure 2K*). Altogether, these data link lymph gland disruption and lamellocyte release into circulation to successful wasp egg encapsulation.

## Fighting wasp parasitism requires Toll/NF-κB activation in PSC cells

Having established that Dif plays an essential role in the lymph gland response to wasp infection, we analyzed whether and in which lymph gland cells Dif activity is required. Immunostaining of *wt* lymph glands shows that Dif is expressed in PSC cells (*Figure 3A–A''*) (*Gueguen et al., 2013*). 6 hr post-parasitism, it is also expressed in hematopoietic progenitors (*Figure 3B–B''*). NF-κB transcriptional activity can be monitored by the expression of a reporter transgene, *D4-LacZ*, which consists of four tandemly repeated NF-κB-binding sites upstream of *lacZ* (*Flores-Saaib et al., 2001*). In agreement with published data, *D4–lacZ* is expressed in PSC cells in uninfected *wt* larvae and upregulated in these cells 6 hr post parasitism, in addition to being expressed de novo in lymph gland progenitors (*Figure 3C–C'', D–D'', E*) (*Gueguen et al., 2013*). In *pll[2]/pll[7]* mutants, *D4-LacZ* expression is still observed in the PSC under normal conditions (*Figure 3F–F''*), whereas it is not expressed in the PSC in *Dif[1]* mutants (*Figure 3I–I''*). This indicates a Toll-independent, basal activity of Dif. 6 hr post-parasitism, no upregulation of *D4-LacZ* occurs in PSC cells in either *pll[2]/pll[7]* or *Dif[1]* mutants (*Figure 3G–G'', J–J''*, quantification in *Figure 3H*). These data show that Toll/NF-κB signaling is activated in the PSC upon parasitism and that this activation requires *Dif*. Further validating this conclusion, we found that *D4-LacZ* is not upregulated 6 hr post-parasitism when *Dif* is specifically down-regulated in PSC cells by driving *Dif* dsRNA expression with the PSC driver p*col-Gal4* (*col > Dif* RNAi) (*Figure 3K,L*; *Figure 3—figure supplement 1A,B*). 6 hr post-parasitism, unlike in *pll[2]/pll[7]* mutants (*Figure 3G*), low levels of *D4-LacZ* expression persist in *Dif[1]* mutant lymph gland progenitors

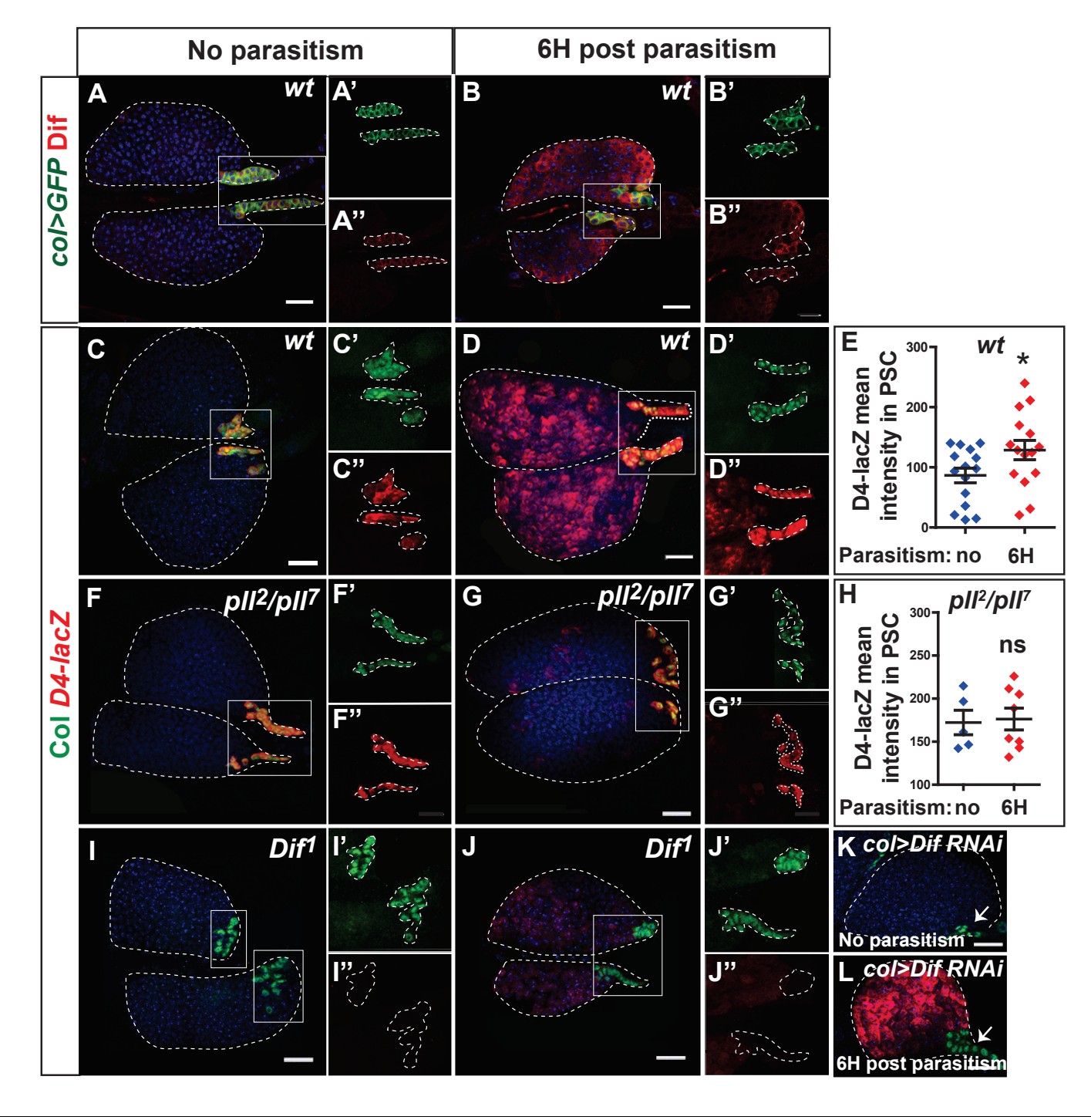

**Figure 3.** Dif-dependent Toll/NF-kB activation in PSC cells. (A, B) Dif (red) immunostaining of *col > mCD8* GFP (*col > GFP*, green) lymph glands without (A) and 6 hr post-parasitism (B). (A', A'', B', B'') enlarged views of the PSC. (C, D, F, G, I, J, K, L) LacZ (red) and Col (green) immunostaining of *D4–LacZ* of *wt* (C, D), *pll²/pll⁷* (F, G), *Dif¹* (I, J), and *col > Dif RNAi* (K, L) lymph glands without (C, F, I, K), and 6 hr post-parasitism (D, G, J, L). (C', C'', D', D'', F', F'', G', G'', I', I'', J' and J'') enlarged views of the PSC. Arrows (K, L) indicate Col-positive PSC cells that do not express *D4–LacZ*. (E, H) Quantifications of *D4-LacZ* mean intensity in PSC cells. Error bars represent SDs, *p<0.1 and ns (not significant) (Mann-Whitney nonparametric test).

DOI: https://doi.org/10.7554/eLife.25496.014

The following source data and figure supplement are available for figure 3:

**Source data 1.** D4-lacZ staining quantification.

DOI: https://doi.org/10.7554/eLife.25496.016

*Figure 3 continued on next page*

*Figure 3 continued*

**Source data 2.** D4-lacZ staining quantification.
DOI: https://doi.org/10.7554/eLife.25496.017
**Figure supplement 1.** Dif, Dorsal and Drosomycin-GFP expression in lymph gland cells.
DOI: https://doi.org/10.7554/eLife.25496.015

(compare *Figure 3J and D*), possibly reflecting *Dorsal* expression in these cells (*Figure 3—figure supplement 1C,D*) (*Gueguen et al., 2013*). Altogether, these data establish that Toll/NF-κB signaling is induced both in the PSC and in lymph gland progenitors in response to parasitism and that Dif is the major NF-κB transactivator involved.

In order to distinguish between possible roles of Toll/NF-κB signaling in the PSC and progenitors following parasitism, we knocked down different components of the pathway, such as *Dif* (*Figure 3—figure supplement 1A,B*), *pll* and *Myd88*, by expressing RNAi in either PSC cells (*col>*) or in lymph gland progenitors using *domeless-Gal4* (*dome>*). Toll/NF-κB down-regulation in MZ progenitors (*dome > Myd88* RNAi; *dome > cact > Dif* RNAi) neither significantly reduced lymph gland dispersal (*Figure 4—figure supplement 1C,D*) nor lymph gland lamellocyte differentiation 20 hr post-parasitism (*Figure 4—figure supplement 2A,B*). By contrast, downregulation of Toll/NF-κB signaling in PSC cells (*col > Dif* RNAi; *col > Myd88* RNAi; *col > pll* RNAi) delayed lymph gland dispersal post-parasitism (*Figure 4A*). Similar results were obtained when *antp-Gal4* (*antp>*), another PSC driver, was used (*Figure 4—figure supplement 1A,B*). Next, we analyzed lymph gland lamellocyte differentiation 20 hr post-parasitism. Whereas lamellocytes were detected in control lymph glands, few, if any were found when Toll/NF-κB signaling was impaired in PSC cells (*Figure 4B,C*), corroborating our previous interpretation that lymph gland disruption is dependent on lamellocyte differentiation (*Figure 1*). Finally, we determined whether Toll/NF-κB signaling in PSC cells is required for successful wasp egg encapsulation. Whereas only 20% of wasp eggs hatched in control larvae (*col>*), 70% and 80% hatched when *Myd88* or *Dif* were down-regulated in PSC cells (*col > Myd88* RNAi; *col > Dif* RNAi), respectively (*Figure 4D*). Altogether, these data show that Toll/NF-κB signaling is required in PSC cells and not in lymph gland progenitors for the correct timing of lymph gland lamellocyte differentiation, which leads to lymph gland dispersal and thus a successful cellular immune response against wasp parasitism.

## Spätzle and SPE are required for Toll/NF-kB activation

Spätzle (Spz), the ligand of the Toll/NF-κB pathway, is synthesized and secreted as an inactive cytokine, and its activation depends on cleavage by the Spätzle Processing Enzyme (SPE) (*Jang et al., 2006*). Previous studies established that both Spz and SPE are essential for wasp egg encapsulation (*Paddibhatla et al., 2010*; *Sorrentino et al., 2004*). Here, we further show that *spz* null mutant (*spz^rm7*) larvae exhibit a strong lymph gland dispersal delay (*Figure 5A*). Furthermore, decreasing SPE expression in PSC cells (*col > SPE* RNAi) severely delays lymph gland dispersal (*Figure 5B*), whereas knocking down its expression in progenitors (*dome > SPE* RNAi) has no significant effect (*Figure 5C*). This suggests that SPE expression in PSC cells is required for Toll/NF-κB activation in the lymph gland. To confirm this, we analyzed *D4-lacZ* expression in *col > SPE* RNAi larvae. 6 hr post-parasitism, in contrast to control larvae, *D4-LacZ* expression is not upregulated either in the PSC or in progenitors (*Figure 5D–F*). Altogether, these data indicate that, in response to parasitism, SPE expression in PSC cells is essential for the production of active Spz and for Toll/NF-κB activation in all lymph gland cells.

SPE itself is activated by proteolytic cleavage (*Jang et al., 2006*). Two parallel extracellular cascades, namely a pattern recognition receptor-dependent pathway involving the protease ModSP (Modular Serine Protease), and a Psh (Persephone)-dependent pathway are involved in SPE processing upon bacterial and fungal infections (reviewed in [*Valanne et al., 2011*]). To address whether ModSP and/or Psh are required in the fight against wasp parasitism, we looked at wasp egg hatching in mutants (*Figure 5—figure supplement 1A*). No significant difference, compared to the control, was observed in *modSP^1* mutant, whereas survival of wasp larvae was significantly enhanced in *psh^1* and *psh^1; modSP^1* mutant larvae, indicating that Psh is required for successful wasp egg encapsulation. We further observed that lymph gland disruption is delayed in *psh^1* mutant larvae (*Figure 5—figure supplement 1B*), demonstrating that Psh is required for timely lymph gland dispersal.

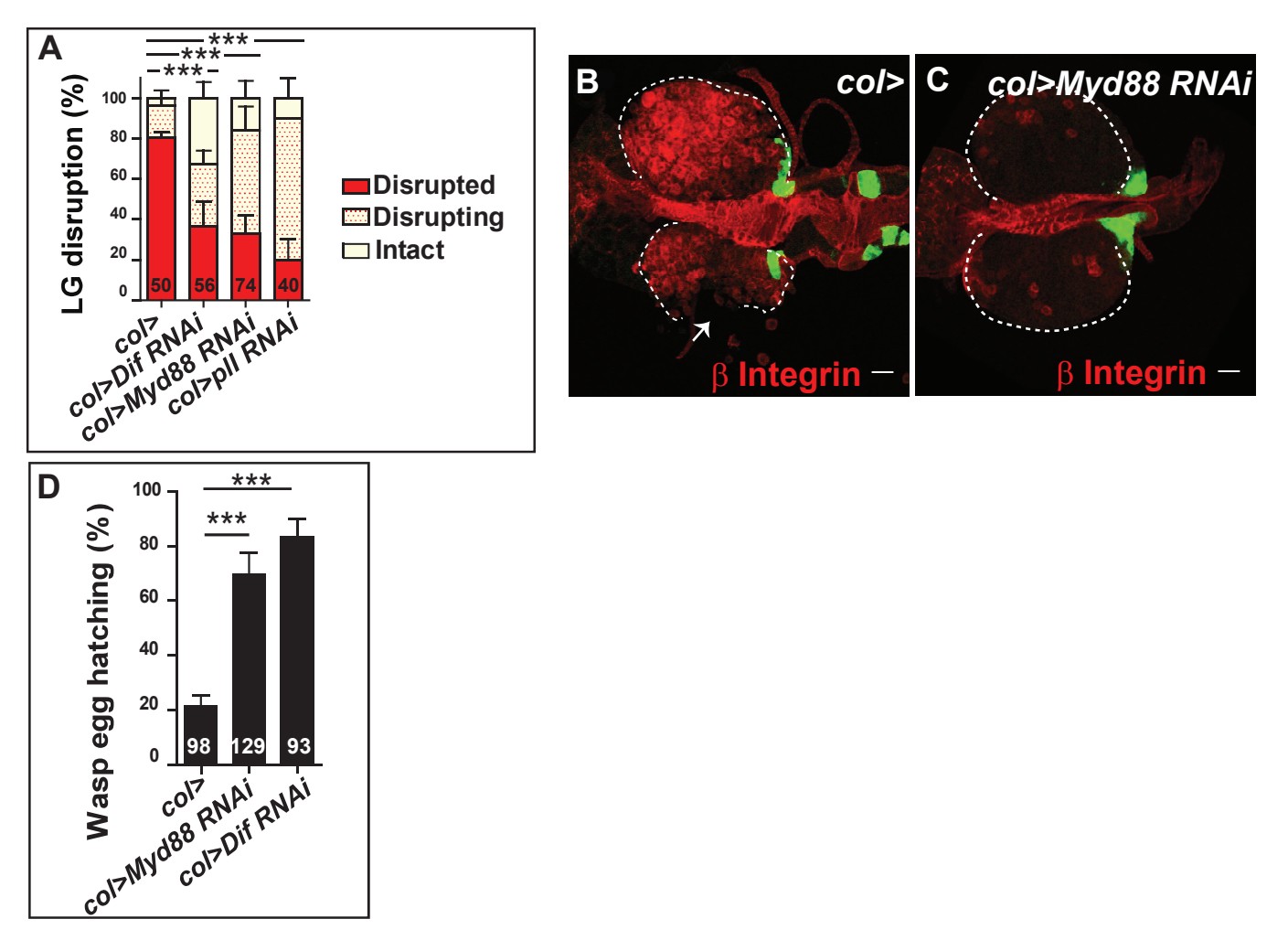

**Figure 4.** Toll/NF-kB activation in PSC cells controls lymph gland dispersal, lamellocyte differentiation and wasp egg encapsulation. (**A**) Quantification (%) of lymph gland disruption post parasitism in *col>* (control), *col > Dif RNAi*, *col > Myd88 RNAi* and *col > pll RNAi* lymph glands. (**B–C**) Representative confocal images of lamellocyte differentiation 20 hr post-parasitism (β Integrin, red) in control lymph glands (*col > GFP*) (**B**) and when Toll/NF-kB is downregulated in PSC cells (*col > GFP > Myd88 RNAi*) (**C**). At least 60% (n = 44) of lymph glands displayed numerous lamellocytes in the control, whereas less than 15% (n = 41) are observed in *col > Myd88 RNAi* lymph glands. Note early signs of lymph gland disruption in (B, arrow). (**D**) Quantification (%) of wasp egg hatching 48 hr post-parasitism in *col>* (control), *col > Myd88 RNAi* and *col > Dif RNAi* lymph glands. Error bars correspond to SEM, ***p<0.001 (Pearson's Chi-squared test).

DOI: https://doi.org/10.7554/eLife.25496.018

The following source data and figure supplements are available for figure 4:

**Source data 1.** Lymph gland disruption quantification.
DOI: https://doi.org/10.7554/eLife.25496.022
**Source data 2.** Wasp egg hatching quantification.
DOI: https://doi.org/10.7554/eLife.25496.023
**Figure supplement 1.** Toll/NF-kB activation in PSC cells but not in lymph gland progenitors is required for proper lymph gland disruption in response to wasp parasitism.
DOI: https://doi.org/10.7554/eLife.25496.019
**Figure supplement 1—source data 1.** Lymph gland disruption quantification.
DOI: https://doi.org/10.7554/eLife.25496.020
**Figure supplement 2.** Toll/NF-kB signaling is not required in lymph gland progenitors for lamellocyte differentiation.
DOI: https://doi.org/10.7554/eLife.25496.021

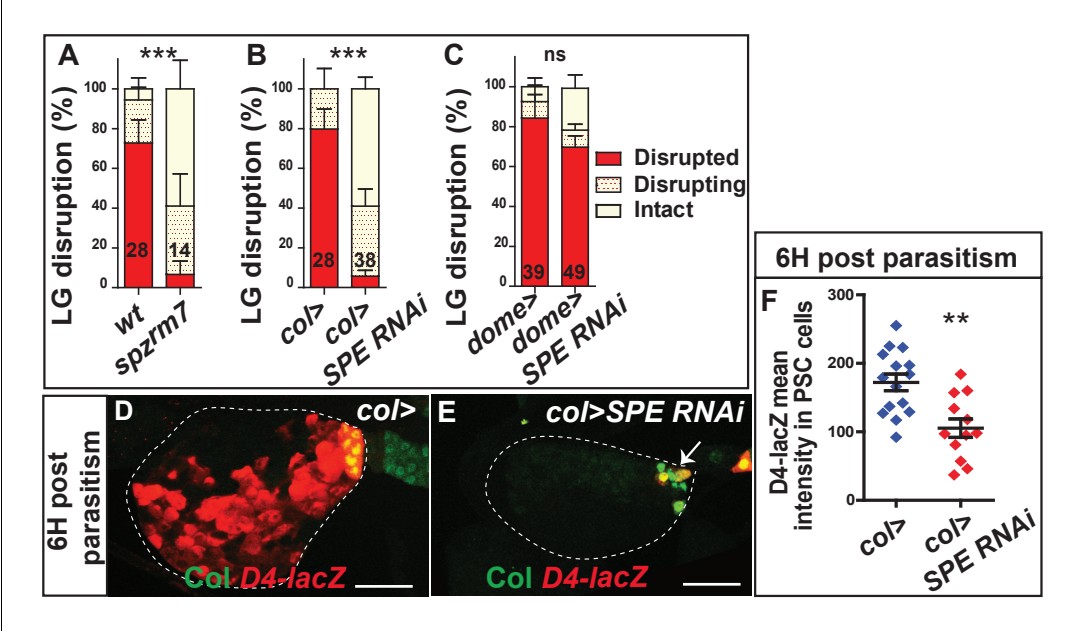

**Figure 5.** Toll/NF-kB activation depends on SPE expression in PSC cells. (A–C) Quantification (%) of lymph gland disruption post parasitism in $spz^{rm7}$ larvae (A) and when SPE is down-regulated in either PSC cells (*col > SPE RNAi*) (B) or lymph gland progenitors (*dome > SPE RNAi*) (C). Error bars correspond to SEM, ***p<0.001 and ns (not significant) (Pearson's Chi-squared test). (D, E) LacZ (red) and Col immunostaining (green) of *D4–LacZ* expressing lymph glands 6 hr post-parasitism in *wt* (*col>*) (D) and when SPE is down-regulated in the PSC (white arrow) (*col > SPE RNAi*) (E). (F) Quantifications of *D4-lacZ* mean intensity in PSC cells 6 hr post-parasitism. Error bars represent SDs. **p<0.01 (Mann-Whitney nonparametric test).
DOI: https://doi.org/10.7554/eLife.25496.024

The following source data and figure supplements are available for figure 5:

**Source data 1.** Lymph gland disruption quantification.
DOI: https://doi.org/10.7554/eLife.25496.028
**Source data 2.** Lymph gland disruption quantification.
DOI: https://doi.org/10.7554/eLife.25496.029
**Source data 3.** Lymph gland disruption quantification.
DOI: https://doi.org/10.7554/eLife.25496.030
**Figure supplement 1.** Psh is required for wasp egg neutralization and timely lymph gland disruption.
DOI: https://doi.org/10.7554/eLife.25496.025
**Figure supplement 1—source data 1.** Wasp egg hatching quantification.
DOI: https://doi.org/10.7554/eLife.25496.026
**Figure supplement 1—source data 2.** Lymph gland disruption quantification.
DOI: https://doi.org/10.7554/eLife.25496.027

These results strongly suggest that Psh-dependent Spz cleavage activates the Toll/NF-κB pathway in lymph gland cells following parasitism.

## Toll/NF-κB activation in PSC cells upon parasitism requires high ROS levels

ROS levels are increased in PSC cells in response to wasp parasitism (*Sinenko et al., 2011*). We therefore investigated possible cross-regulation between ROS and Toll/NF-κB signaling. To monitor ROS levels, we looked at the expression of the *gstD-lacZ* transgene, where *lacZ* is placed under the control of the promoter of *Glutathione S-transferase D1*, which encodes an anti-oxidant enzyme that is expressed in response to oxidative stress (*Sykiotis and Bohmann, 2008*). In the absence of parasitism, low, scattered expression of *gstD-lacZ* was observed in the lymph gland (*Figure 6A*), consistent with a previous report (*Owusu-Ansah and Banerjee, 2009*). 6 hr post-parasitism, *gstD-lacZ* expression is increased in both the PSC (*Figure 6B,C*) (*Sinenko et al., 2011*) and the progenitors (*Figure 6B,D*), indicating that parasitism induces a global oxidative stress in the lymph gland.

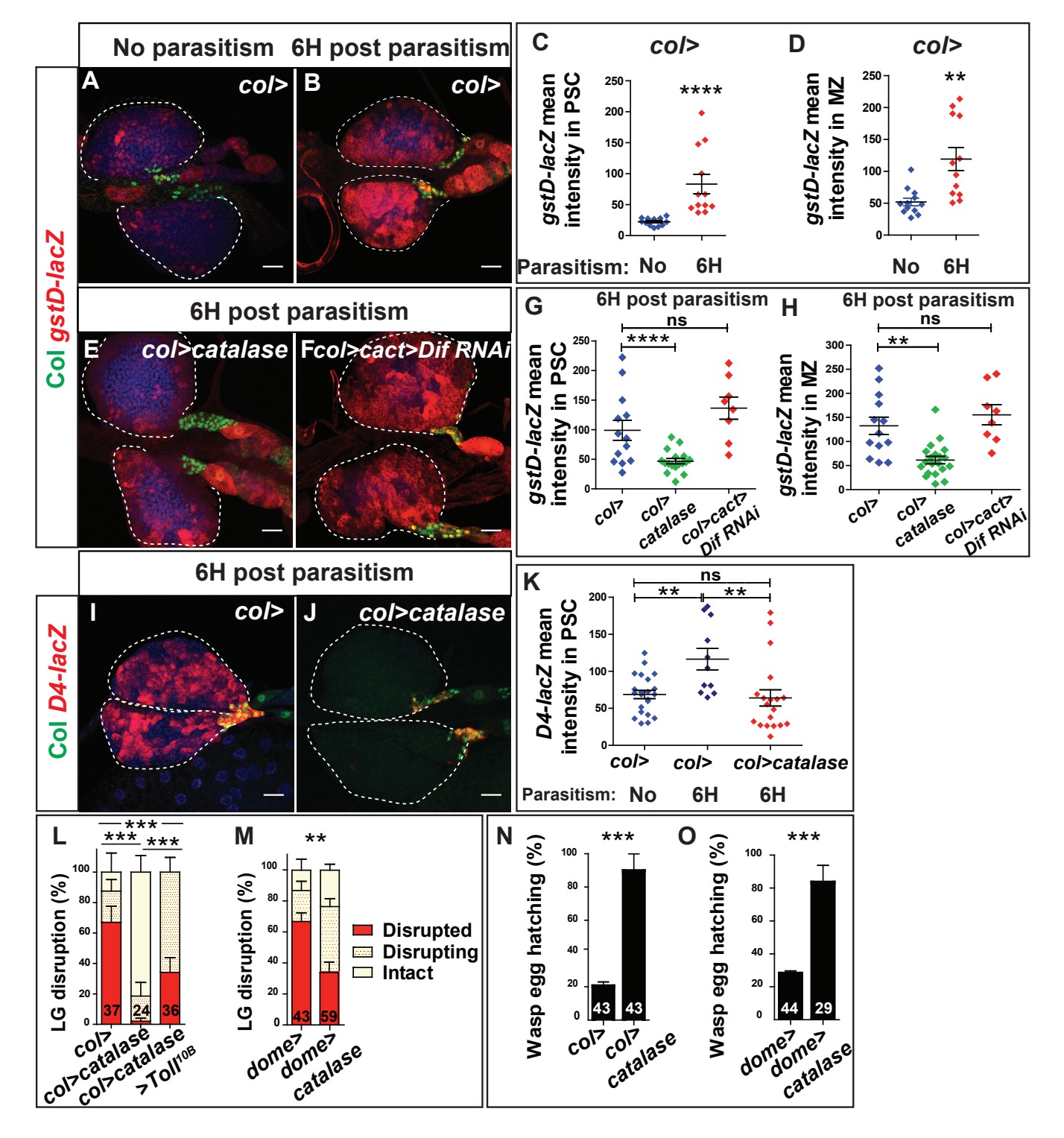

**Figure 6.** Wasp-mediated oxidative stress promotes Toll/NF-kB activation in the PSC and lymph gland disruption. (A, B, E, F) LacZ (red) and Col (green) immunostaining in *gstD-lacZ* expressing lymph glands in *wt* context (*col>*; A, B), when Catalase is expressed in the PSC (*col > catalase*) (E) and when Toll/NF-kB is inactivated in PSC cells (*col > cact > Dif RNAi*)(F). (C, D, G, H) Quantifications of *gstD-lacZ* mean intensity in PSC and progenitor (MZ) cells. Error bars represent SDs. **p<0.01, ****p<0.0001 and ns (not significant) (Mann-Whitney nonparametric test). (I, J) LacZ (red) and Col (green) immunostaining in *D4-LacZ* expressing lymph glands in *wt* (*col>*) (I) and when an intracellular catalase is expressed in the PSC (*col > catalase*) (J). (K) Quantifications of *D4-lacZ* mean intensity in PSC cells. **p<0.01 and ns (not significant) (Mann-Whitney nonparametric test). (L–M) Quantifications of
*Figure 6 continued on next page*

*Figure 6 continued*

lymph gland disruption post parasitism. (**N,O**) Quantification (%) of wasp egg hatching. Error bars correspond to SEM, **p<0.01, ***p<0.001 (Pearson's Chi-squared test).

DOI: https://doi.org/10.7554/eLife.25496.031

The following source data and figure supplements are available for figure 6:

**Source data 1.** gstD-lacZ staining quantification.
DOI: https://doi.org/10.7554/eLife.25496.035
**Source data 2.** D4-lacZ staining quantification.
DOI: https://doi.org/10.7554/eLife.25496.036
**Source data 3.** Lymph gland disruption quantification.
DOI: https://doi.org/10.7554/eLife.25496.037
**Source data 4.** Lymph gland disruption quantification.
DOI: https://doi.org/10.7554/eLife.25496.038
**Source data 5.** Wasp egg hatching quantification.
DOI: https://doi.org/10.7554/eLife.25496.039
**Figure supplement 1.** Duox- and Nox-independent generation of ROS in PSC cells 6 hr post-parasitism.
DOI: https://doi.org/10.7554/eLife.25496.032
**Figure supplement 2.** ROS are required both in PSC cells and lymph gland progenitors for lamellocyte differentiation.
DOI: https://doi.org/10.7554/eLife.25496.033
**Figure supplement 3.** Scavenging ROS in PSC cells or silencing the EGFR pathway in MZ progenitors delays lymph gland disruption upon parasitism.
DOI: https://doi.org/10.7554/eLife.25496.034

Interestingly, depleting ROS specifically in PSC cells by expressing an intracellular anti-oxidant catalase (*col > catalase*) strongly decreased *gstD-lacZ* expression in both the PSC and progenitors (compare *Figure 6E and B*, quantified in G, H). In contrast, expressing the extracellular catalase IRC (Immune-Related Catalase) (*Ha et al., 2005*) in PSC cells (*col > IRC*), did not prevent *gstD-lacZ* expression in the lymph gland (*Figure 6—figure supplement 1D*). Likewise, PSC downregulation of the membrane NADPH oxidase enzymes Duox (dual oxydase) (*col > Duox* RNAi) or Nox (*col > Nox* RNAi) (*Bae et al., 2010*), able to generate extracellular ROS, did not prevent *gstD-lacZ* upregulation in PSC cells (*Figure 6—figure supplement 1B,C*). Altogether, these results indicate that following parasitism, ROS are intrinsically produced by PSC cells, which in turn leads to oxidative stress in lymph gland progenitors. Switching off the Toll/NF-κB pathway in PSC cells (*col > cact > Dif RNAi*) has no effect on *gstD-lacZ* expression 6 hr post-parasitism (compare *Figure 6F and B*, quantified in G and H), indicating that the increase of ROS levels does not require Toll/NF-κB activity. In contrast, scavenging ROS in PSC cells both suppresses *D4-lacZ* expression in progenitors and its upregulation in the PSC (*Figure 6I–K*), leading us to conclude that Toll/NF-κB activation in PSC cells is a response to increased ROS levels in these cells. To confirm the epistatic relationship between ROS and Toll/NF-κB in PSC cells, we simultaneously reduced ROS levels and activated Toll/NF-κB, through the expression of a constitutively active form of the Toll receptor (*Toll[10B]*) in PSC cells (*col > catalase > Toll[10B]*). A partial rescue of lymph gland dispersal was observed (*Figure 6L*). These data indicate that, in response to wasp egg laying, ROS level increase in PSC cells leads to Toll/NF-κB activation and lymph gland dispersal.

Since 6 hr post parasitism *gstD-lacZ* is expressed in all lymph gland cells, we examined in which cells modifying ROS levels could affect the immune response. Decreasing ROS levels either in the PSC (*col > catalase*) or in progenitors (*dome > catalase*) delay lymph gland lamellocyte differentiation (*Figure 6—figure supplement 2*), postpone lymph gland rupture (*Figure 6L,M*; *Figure 6—figure supplement 3A–C*) and enhance wasp egg hatching (*Figure 6N,O*). Thus, increased ROS levels in both PSC cells and progenitors following parasitism are essential for the mounting of an efficient lymph gland response and for wasp egg encapsulation.

## Toll/NF-κB and EGFR/Erk pathways act in parallel to trigger lymph gland disruption

The incomplete rescue of lymph gland dispersal when simultaneously scavenging ROS and activating Toll/NF-κB signaling in PSC cells, suggested that additional signaling pathway(s) act downstream of ROS. One obvious candidate is EGFR/Erk which is required for wasp egg encapsulation

(*Sinenko et al., 2011*). According to the authors, increased ROS levels in PSC cells, in response to wasp parasitism, lead to the secretion of Spitz (sSpi) into the hemolymph, which activates EGFR/Erk signaling in circulating hemocytes and triggers their differentiation into lamellocytes. However, a direct role of EGFR activation in lymph gland progenitors was not ruled out. We therefore looked at phosphorylated ERK (p-ERK), an EGFR activation readout (*Sinenko et al., 2011*). Whereas p-ERK was not, or barely, detected in the lymph gland under normal conditions (*Figure 7A–A''*), it was detected in lymph gland progenitors 6 hr post parasitism, indicating EGFR activation (*Figure 7B–B''*). Furthermore, decreasing *Spitz* expression in the PSC (*antp >spi* RNAi) or down-regulating the EGFR pathway in lymph gland progenitors, by expressing a dominant-negative form of the EGFR receptor (*dome > Egfr^{DN}*), delays lymph gland dispersal post-parasitism (*Figure 7C,D*; *Figure 6— figure supplement 3D–F*). Collectively, these data indicate that, in response to parasitism, the EGFR pathway is activated in lymph gland progenitors, leading to lymph gland dispersal.

For proper timing of lymph gland dispersal in response to parasitism, EGFR and Toll/NF-κB pathways are required in progenitors and PSC cells, respectively. This raised the question of whether

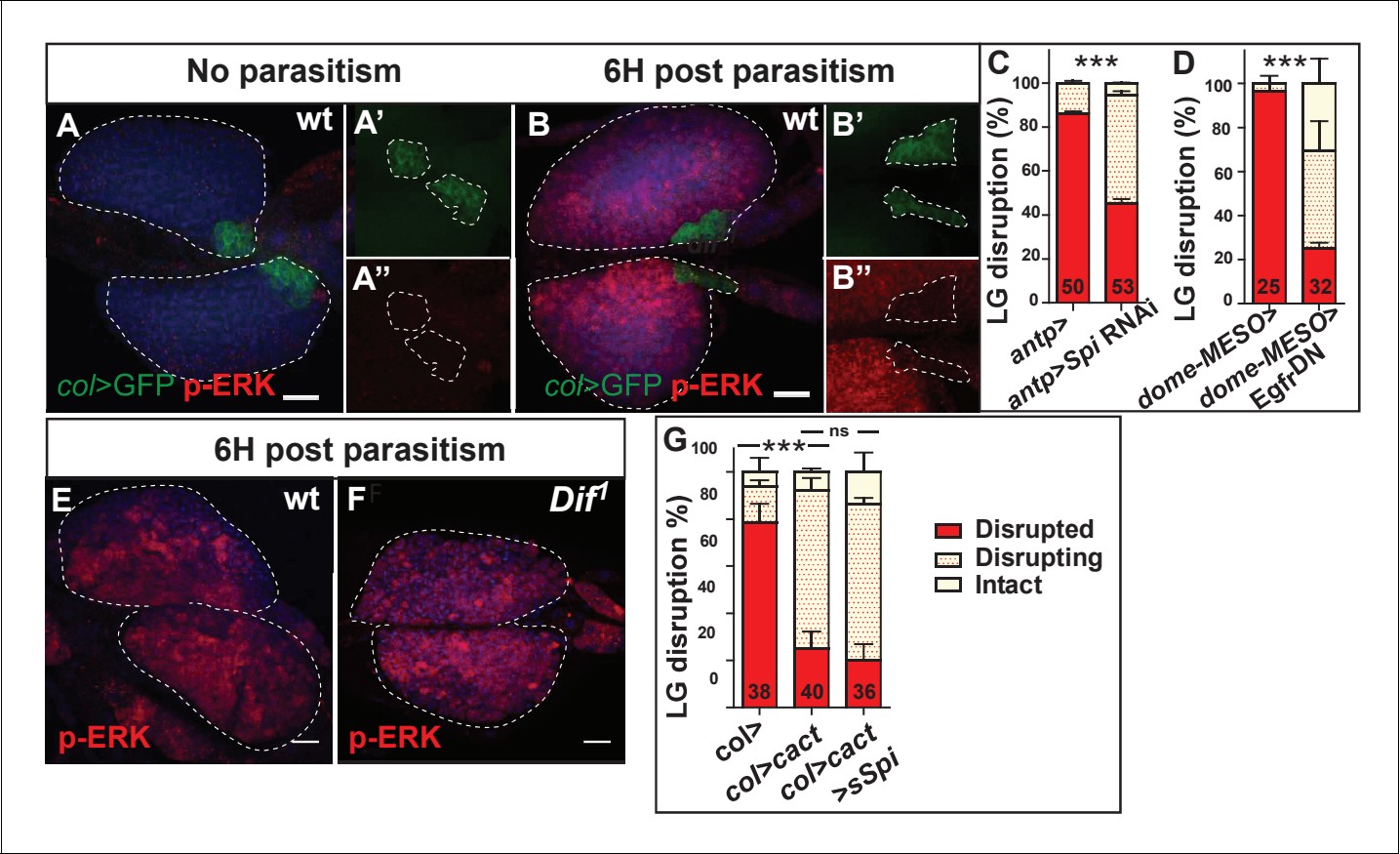

**Figure 7.** Epistatic relationships between EGFR/Erk and Toll/NF-κB signaling. (A, B) p-ERK (red) immunostaining in *col > GFP* (PSC, green) larvae. (A′, A″, B′, B″), enlarged views of PSC cells. (C, D) Quantifications of lymph gland disruption post parasitism. (E, F) p-ERK (red) immunostaining in *wt* (E) and *Dif^1* (F) mutant lymph glands. (G) Quantifications (%) of lymph gland disruption post parasitism. Error bars correspond to SEM, ***p<0.001, ns (not significant) (Pearson's Chi-squared test).

DOI: https://doi.org/10.7554/eLife.25496.040

The following source data is available for figure 7:

**Source data 1.** Lymph gland disruption quantification.
DOI: https://doi.org/10.7554/eLife.25496.041
**Source data 2.** Lymph gland disruption quantification.
DOI: https://doi.org/10.7554/eLife.25496.042
**Source data 3.** Lymph gland disruption quantification.
DOI: https://doi.org/10.7554/eLife.25496.043

these two pathways are functionally connected. We therefore tested whether Toll/NF-κB activation in PSC cells controls *Spitz* expression, by looking at p-ERK expression in progenitors of parasitized *Dif*[1] mutant larvae. No significant difference with controls was observed (*Figure 7E,F*), indicating that Toll/NF-κB activation is not required for EGFR activation. Second, we reciprocally tested whether expressing sSpi in PSC cells could rescue the lymph gland dispersal delay observed upon Toll/NF-κB inactivation in the PSC. Simultaneous overexpression of sSpi and inactivation of Toll/NF-κB in PSC cells (*col > cactus > sSpi*) did not rescue the lymph gland dispersal defect (*Figure 7G*), leading us to conclude that Toll/NF-κB and EGFR pathways act in parallel.

## Discussion

Toll/NF-κB signaling is well known in *Drosophila* for its roles in embryonic dorsal/ventral axis formation and in the humoral innate immune response (*Lemaitre and Hoffmann, 2007*; *Moussian and Roth, 2005*). In mammals, it plays important roles in various aspects of innate and adaptive immunity (*Espín-Palazón and Traver, 2016*). Toll/NF-κB function in the *Drosophila* cellular response induced by wasp parasitism, although long reported (*Qiu et al., 1998*; *Sorrentino et al., 2004*), has only recently started to be investigated in some detail (*Gueguen et al., 2013*; *Schmid et al., 2014*). Inactivation of the pathway in two immune tissues, namely the fat body and hemocytes, did not impact on either wasp egg encapsulation or lymph gland lamellocyte differentiation and disruption (*Schmid et al., 2014*) (data not shown). Thus, when and in which cells Toll/NF-κB signaling is activated and required to fight wasp infection remained unknown. Here, we provide evidence that in response to *Leptopilina boulardi* parasitism, activation of Toll/NF-κB signaling in PSC cells non cell-autonomously controls the timing of wasp-induced lymph gland dispersal and the release of lymph gland lamellocytes into circulation, both required to prevent wasp egg hatching. Toll/NF-κB activation in PSC cells is dependent on both an increase in ROS levels in these cells, and the expression of SPE which cleaves the ligand Spz. Our data show that EGFR pathway activation in lymph gland progenitors is also required for on time lymph gland dispersal. This activation is mediated by the secretion of Spitz from PSC cells, itself dependent upon ROS signaling. In conclusion, our data identify a

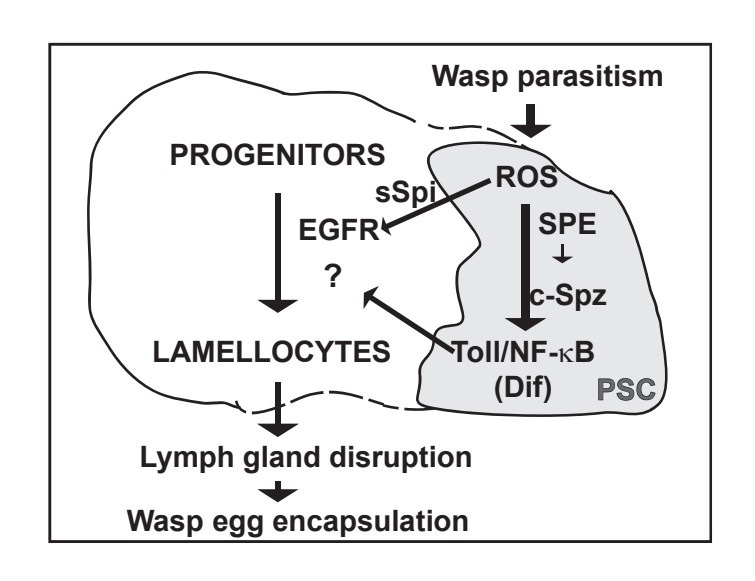

**Figure 8.** Proposed gene regulatory network that controls lymph gland rupture upon wasp parasitism. The PSC is drawn in grey. Wasp parasitism increases ROS in PSC cells that activate Toll/NF-κB and Spitz secretion (sSpi). Toll/NF-kB activation in PSC cells requires SPE in the same cells for Spätzle processing (c-Spz). sSpi non cell-autonomously activates the EGFR pathway in lymph gland progenitors. Both EGFR and Toll/NF-κB activation are required for lymph gland lamellocyte differentiation, lymph gland disruption and wasp egg encapsulation.
DOI: https://doi.org/10.7554/eLife.25496.044

signaling cascade that links the PSC response to wasp parasitism to lymph gland lamellocyte differentiation, lymph gland disruption and wasp egg neutralization (*Figure 8*).

In the absence of parasitism, hemocyte homeostasis in the lymph gland relies on various signaling pathways active in progenitors and/or differentiating hemocytes, including Hedgehog, Wnt, Pvf and JAK/STAT (for review see *Letourneau et al., 2016*). Previous work established that downregulation of Col expression or JAK/STAT signaling in progenitors, is a prerequisite for wasp-induced lamellocyte differentiation in the lymph gland (*Makki et al., 2010*; *Oyallon et al., 2016*). Our present data show that increasing ROS levels in PSC cells, in response to parasitism, leads to activation of Toll/NF-κB and EGFR signaling in the PSC and hematopoietic lymph gland progenitors, respectively, and that both signaling events are required for timely lamellocyte differentiation. How EGFR signaling, a signal downstream of NF-κB integrates with other signaling pathways active in hematopoietic progenitors, remains to be addressed.

Surviving wasp parasitism is dependent on the ability of the *Drosophila* larva to reroute basal hematopoiesis and produce, in a timely manner, lamellocytes to neutralize wasp eggs before they hatch in the fly larva. Here, we establish a critical function of Toll/NF-κB signaling in PSC cells in ensuring the proper timing of lamellocyte differentiation and lymph gland disruption in response to parasitism. This temporal control is essential for a successful neutralization of the wasp egg. *Drosophila* lamellocytes have been reported to differentiate both from sessile/circulating hemocytes of embryonic origin, and from larval lymph gland progenitors (*Anderl et al., 2016*; *Crozatier et al., 2004*; *Honti et al., 2010*; *Lanot et al., 2001*; *Márkus et al., 2009*; *Sorrentino et al., 2002*; *Stofanko et al., 2010*). This dual origin poses questions as to the relative contributions of these two lamellocyte sources to successful wasp egg encapsulation. We found that, while few lamellocytes are found in circulation before lymph gland disruption, after disruption, their number sharply increases, constituting nearly half of all circulating hemocytes. We further observed that delaying lymph gland disruption impairs the success of wasp egg encapsulation. Altogether, these data indicate that the release of lymph gland lamellocytes is paramount for successful wasp egg neutralisation. One independent study concluded that the sessile compartment is the major source of lamellocytes necessary for fighting parasitism, although melanisation of wasp eggs rather than their hatching was examined (*Márkus et al., 2009*). In addition, the authors reported a substantial delay of lymph gland disruption (i.e. 48 hr post-wasp egg deposition) in their control condition, compared to 20 hr in our study. The discrepancy between their and our present conclusions could therefore, in part, reflect the use of different conditions for wasp infestation, including the time and/or the temperature since the time scale of lymph gland disruption is highly dependent on temperature (*Figure 2—figure supplement 2*), or the use of distinct fly genetic backgrounds (*Sorrentino et al., 2004*). Here, we have investigated the success of wasp egg hatching in different conditions, and our data establish a functional link between timely differentiation of lamellocytes in the lymph gland, their release into circulation and efficient wasp egg neutralization.

In *Drosophila*, Gram-positive or fungal infections trigger the activation of the Toll/NF-κB pathway, leading to the systemic production of antimicrobial peptides by the fat body, a functional equivalent of the mammalian liver (*Lemaitre and Hoffmann, 2007*). The antifungal peptide Drosomycin, one main effector of the Toll humoral response, is directly regulated by NF-κB (*Ferrandon et al., 1998*). We have shown here that wasp parasitism activates Toll/NF-κB in the lymph gland. Drosomycin-GFP expression was, however, not observed in response to wasp parasitism in the lymph gland (*Figure 3—figure supplement 1E,F*), in agreement with previous observations that antimicrobial peptides are not produced in larvae under these conditions (*Nicolas et al., 1996*). This suggests that Dif target genes activated in the lymph gland under parasitism are specific to this response. Among these targets, identifying which one(s) code for signal(s) required to trigger efficient lamellocyte differentiation in the lymph gland is of primary interest. Indeed, we found that it directly impacts the timing of lymph gland dispersal. A recent study established that constitutive activation of Toll/NF-κB in PSC cells impairs the permeability barrier of the PSC, possibly facilitating diffusion of signal(s) from the PSC to lymph gland progenitors (*Khadilkar et al., 2017*). Whether modification of the permeability barrier is also triggered by Toll/NF-κB signaling in PSC cells in response to parasitism to promote lamellocyte differentiation in the lymph gland remains to be investigated.

Oxidative stress, in particular reactive oxygen species (ROS), regulates hematopoiesis both in mammals and in *Drosophila* (*Ludin et al., 2014*; *Owusu-Ansah and Banerjee, 2009*; *Sinenko et al., 2011*). In mammals, there is a complex interplay between ROS and NF-κB signaling during this process.

Oxidative stress can either affect IκB kinase complex (IKK) activity in the cytoplasm or NF-κB DNA-binding capacity in the nucleus (*Gupta et al., 2014*; *Siomek, 2012*). In the fly lymph gland, increased ROS levels are observed in both the PSC and hematopoietic progenitors in response to parasitism. ROS increase in lymph gland progenitors, which could reflect changes in cell metabolism since progenitors actively divide as an immediate response to parasitism (*Krzemien et al., 2010b*; *Sorrentino et al., 2002*), is dependent on ROS production by PSC cells. ROS-induced signaling in both PSC cells and progenitors is required to trigger wasp-induced dispersal of the lymph gland and release of lamellocytes into circulation. This ROS effect is both relayed by Toll/NF-κB activation in PSC cells and EGFR activation in lymph gland progenitors (*Sinenko et al., 2011*). ROS-dependent Toll/NF-κB activation, following infection by the *Wolbachia* bacteria, was reported in the mosquito *Aedes aegypti,* but the mechanism involved was not defined (*Pan et al., 2012*). How can an oxidative burst in PSC cells lead to Toll/NF-κB activation in the same cells? ROS could control the production by PSC cells of secreted product(s) required for Spätzle activation, like SPE (*Figure 5*). Alternatively, Spätzle activation could be independent of the oxidative stress in PSC cells, and ROS could regulate intracellular steps of Toll/NF-κB activation, like the DNA-binding properties of NF-κB, as previously reported in mammals (*Gloire et al., 2006*). What type of regulation is operational remains to be determined.

In mammals, systemic bacterial infection induces an 'emergency granulopoiesis' characterized by de novo production of neutrophils in the bone marrow (*Zhao and Baltimore, 2015*). In this process, the TLR (Toll-like Receptors)/NF-κB pathway is activated via TLR4 in mouse bone marrow endothelial cells, a component of the vascular niche (*Boettcher et al., 2014*). The cellular immune response to parasitism in *Drosophila* can be compared to an emergency hematopoiesis. Establishing that Toll/NF-κB activation in the *Drosophila* hematopoietic niche is essential for this response extends the evolutionary parallels between *Drosophila* and mouse to the control of stress-induced hematopoiesis. The role of the EGFR pathway in mammalian hematopoiesis is largely unknown (*Kerpedjieva et al., 2012*). It will be important to determine whether the regulatory network uncovered here in *Drosophila*, in which ROS dually regulate EGFR and Toll/NF-κB signaling, also operates in vertebrates under emergency hematopoiesis.

# Materials and methods

## Fly strains

$w^{1118}$ (wild type, *wt*), *pcol-Gal4*, *UAS-mCD8-GFP* and PG125*dome-gal4* (*Jung et al., 2005*; *Krzemień et al., 2007*), *Antp-gal4* (*Mandal et al., 2007*), *Df(2L)J4* (*Meng et al., 1999*); *gstD-lacZ* (*Wang et al., 2003*), *UAS-sSpi* (*Sudarsan et al., 2002*), *UAS-Egfr$^{DN}$* (*Saxena et al., 2014*), *D4-lacZ* on the third chromosome (*Flores-Saaib et al., 2001*). A *D4-lacZ* insertion on the second chromosome, at the integration site *attP*-51B (II), was generated in this study by re-injecting the original *D4/hsp70* construct (*Flores-Saaib et al., 2001*; *Pan and Courey, 1992*). *UAS-Cact* was a gift from S. Govind (The Graduate School and University Center of The City University of New York, New York, USA). The isogenic (iso) strains, $w^{1118}$, *Dif$^1$*, *pll$^2$*, *pll$^{21}$*, *dl$^1$*, *Tl$^{rv1}$*, *Tl$^r$* (*Ferreira et al., 2014*) were kindly given by Luis Teixeira; *modSP$^1$*, *psh$^1$*, double *modSP$^1$*; *psh$^1$* mutant, UAS-Duox RNAi, UAS-Nox RNAi and UAS-IRC by Bruno Lemaitre. Other *Drosophila* strains were obtained from the Bloomington (BL) stock center: *pll$^2$* (BL3111), *pll$^7$* (BL3112), *Dif$^1$* (BL36559), *dl$^1$* (BL3236), *spz$^{rm7}$* (BL55718) (*Tzou et al., 2002*), *UAS-catalase* (BL24621), *UAS-Toll$^{10B}$* (BL58987)(*Schneider et al., 1991*), *UAS-pll RNAi* (BL34733 and BL3577) (*Wu et al., 2015*), *UAS-Dif* RNAi (BL30513), or from the Vienna *Drosophila* RNAi stock center (VDRC): *UAS-spi* RNAi (3922) (*Chen et al., 2016*), *UAS-Myd88 RNAi* (25402) (*Schmid et al., 2014*), *UAS-latran RNAi* (19756 and 100881) (*Kallio et al., 2010*); *UAS-SPE RNAi* (*Jang et al., 2006*). For all RNAi experiments, *UAS-Dicer2* (BL24650) was introduced and *Drosophila* development proceeded at 22°C until L1 stage, before shift to 29°C. Controls correspond to Gal4 drivers crossed with $w^{1118}$.

## Generation of *dome-MESO-Gal4* transgenic lines

The *dome-MESO* sequence from *pCasHs43DomeMESO-lacZ* (*Rivas et al., 2008*) was sub-cloned into the Gal4 pBPGUw (17575, Addgene). The resulting plasmid was used to generate *dome-MESO-Gal4* (*dome-MESO>*) transgenic flies using attP/attB technology (*Bischof et al., 2007*). Two independent *Drosophila* lines were created by integration at the *attP*-51B (II) and *attP*-68A4 (III) sites.

*domeMESO-Gal4 > GFP*-positive cells overlapped with *domeless-Gal4 > GFP* (PG125; [*Krzemień et al., 2007*]) progenitors in lymph gland anterior lobes at L3 stage. At L2 stage, in addition to its expression in progenitors, a weak and transient expression was also detected in a subset of PSC cells.

## Immunostaining

Lymph glands were dissected and processed as previously described (*Krzemień et al., 2007*). Antibodies used were rabbit anti-αPS4 (1/200) (*Krzemień et al., 2007*), mouse anti-Col (1/100) (*Krzemień et al., 2007*), rabbit anti-Dif (1/500, gift from D. Ferrandon), rabbit anti-Trol (1/1000, gift from S. Baumgartner), chicken anti-βgal (1/1000, Abcam), mouse anti-diphosphorylated Erk kinase (pERK) antibody (1/100, Sigma-Aldrich), mouse anti-β Integrin (Developmental Studies Hybridoma Bank, CF.6G11c, 1/100) (*Irving et al., 2005*). Nuclei were labeled with TOPRO3 (Thermo Fisher Scientific, Waltham, USA) or Draq5 (Thermo Fisher Scientific). Images were captured using a Zeiss 710 or a Leica SP8 confocal microscope. Three independent biological replicates were analyzed, and one representative image of at least 20 lymph glands analyzed is shown.

## Wasp parasitism, lymph gland rupture and wasp egg hatching assays

Late second instar or early third instar *Drosophila* larvae raised at 27°C or 29°C (for RNAi experiments) were subjected to parasitism for 1 hr at 22°C by *Leptopilina boulardi* (G486 avirulent strain; [*Russo et al., 1996*]). Larvae were allowed to develop further at 27°C or 29°C before dissection 6, 20, 30, or 48 hr post-parasitism. Quantification of lymph gland dispersal and wasp egg hatching was performed in un-fixed infected larvae. Quantification of lymph gland dispersal was performed 20 hr post-parasitism for the isogenic strains, *modSP¹*, *psh¹* mutants, double *modSP¹; psh¹* mutant, *col>*, *dome>*, *domeMESO>* and *antp>* lines or 30 hr post-parasitism for *wt*, *Dif¹*, *pll²/pll⁷*, *dl¹*, *spz^{rm7}*, +/Df(2L)J4, *dl¹*/Df(2L)J4 and *Dif¹*/Df(2L)J4 mutants. Lymph glands were classified into three groups: disrupted when anterior lobes were absent or rudimentary, disrupting when some cells had escaped the anterior lobes, or intact when the anterior lobe border was regular. The percentage of lymph gland dispersal was calculated by scoring the number of lymph glands in each group, divided by the total number of infected larvae. Wasp egg hatching was analyzed 48 hr post-parasitism and the number of fly larvae containing melanized/unhatched wasp eggs or living hatched wasp larvae, was counted. The % of wasp egg hatching was calculated by scoring the number of living hatched wasp larvae inside the body of the dissected fly larvae, divided by the total number of infected fly larvae. In all experiments, genotypes were analyzed in parallel. Each experiment was repeated independently at least three times, and quantification represents the mean of three independent biological replicates. Graphs and statistical analyses t-test (Mann–Whitney nonparametric test) were performed using GraphPad Prism five software.

## Larval bleeding and hemocyte counting

Individual parasitized larvae were bled on microscope coverslips. Hemolymph samples were dried 10 min, fixed for 10 min in 4% paraformaldehyde in 1XPBS, then stained 2 hr with Phalloidin (1/100; Sigma). Samples were mounted in Vectashield (Vector Laboratories) and examined using a Leica SP8 confocal microscope. In parallel, each larva was examined for lymph gland disruption. At least 10 independent bleedings corresponding to intact or disrupted lymph gland larvae were analyzed for control (*hml>*) and *Dif¹* mutant, and 4 and 3 samples, respectively, were quantified. The percentage of lamellocytes and plasmatocytes was calculated relative to the total number of circulating hemocytes. Lamellocytes are easily distinguishable from plasmatocytes by their elongated shape and large size.

## Quantification of expression intensity

Expression intensities of *D4-lacZ* and *gstD-lacZ* reporters in PSC cells were quantified using Fiji software as described previously (*Morin-Poulard et al., 2016*). Briefly, ROIs (Regions Of Interest) corresponding to PSC cells, labeled by Col, were defined, and the mean intensity of reporter staining by pixel in each ROI was quantified. For quantifying expression intensity in lymph gland progenitors (*gstD-lacZ*), representative areas of reporter staining were defined as ROIs in each lymph gland primary lobe, and the mean intensity by pixel was quantified using Fiji software. For each experiment, middle slides of a Z stack were analyzed. In all experiments, genotypes were analyzed in parallel. Each

experiment was repeated independently at least three times, and one quantification is shown. Graphs and statistical analyses t-test (Mann–Whitney nonparametric test) were performed using GraphPad Prism five software.

## Acknowledgements

We thank S Govind, WJ Lee, A Courey, B Lemaitre, L Teixeira, the Bloomington and Vienna Stock Centers for plasmids and fly stocks; D Ferrandon, I Ando, A Moore and T Trenczek for antibodies; L Bataillé, A Davy, M Meister, C Monod, for critical reading of the manuscript. We are grateful to the Toulouse RIO imaging patform, J Favier, and V Nicolas for fly culture. This work was supported by the CNRS, Ministère de la Recherche (ANR) and Association pour la Recherche sur le Cancer (ARC).

## Additional information

### Funding

| Funder | Grant reference number | Author |
|---|---|---|
| Fondation ARC pour la Recherche sur le Cancer | Graduate Student Fellowship | Isabelle Louradour |
| Centre National de la Recherche Scientifique | | Alain Vincent Michèle Crozatier Nathalie Vanzo |
| Agence Nationale de la Recherche | Bench grant and post-doc fellowship | Michèle Crozatier |

The funders had no role in study design, data collection and interpretation, or the decision to submit the work for publication.

### Author contributions

Isabelle Louradour, Conceptualization, Data curation, Formal analysis, Funding acquisition, Investigation, Visualization, Methodology, Writing—original draft; Anurag Sharma, Formal analysis, Visualization, Methodology; Ismael Morin-Poulard, Formal analysis, Methodology; Manon Letourneau, Methodology; Alain Vincent, Conceptualization, Formal analysis, Funding acquisition, Writing—original draft, Project administration, Writing—review and editing; Michèle Crozatier, Conceptualization, Data curation, Formal analysis, Supervision, Funding acquisition, Investigation, Writing—original draft, Project administration, Writing—review and editing; Nathalie Vanzo, Conceptualization, Data curation, Formal analysis, Supervision, Investigation, Visualization, Methodology, Writing—original draft, Project administration, Writing—review and editing

### Author ORCIDs

Alain Vincent https://orcid.org/0000-0002-2769-7501
Nathalie Vanzo http://orcid.org/0000-0002-6659-0299

### Decision letter and Author response

Decision letter https://doi.org/10.7554/eLife.25496.046
Author response https://doi.org/10.7554/eLife.25496.047

## Additional files

### Supplementary files

• Transparent reporting form
DOI: https://doi.org/10.7554/eLife.25496.045

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
