## [Decision Letter]

Thank you for submitting your article "Reactive oxygen species-dependent Toll/NF-κB activation in the *Drosophila* hematopoietic niche confers immune resistance" for consideration by *eLife*. Your article has been reviewed by four peer reviewers, one of whom, Bruno Lemaitre, is a member of our Board of Reviewing Editors, and the evaluation has been overseen by K VijayRaghavan as the Senior Editor. The following individuals involved in review of your submission have agreed to reveal their identity: Élio Sucena.

The reviewers have discussed the reviews with one another and the Reviewing Editor has drafted a list of essential changes that would make this work attractive for publication in *eLife*. A substantial amount of revision has been required.

Summary:

The manuscript of Louradour et al. reveals a new role of the Toll pathway in the disruption of the *Drosophila* lymph gland (LG) and the differentiation of lamellocytes from the LG. It provides new insights on the mechanism of encapsulation, which is an important immune defense mechanism against parasitoid wasps. The main conclusion is that a wasp infection triggers a burst of ROS in the posterior signaling center (PSC) of the lymph gland, which results in Toll activation in the PSC and in hemocyte progenitors of the LG. It additionally leads to the production of Spitz, a ligand of the EGFR-pathway. Both pathways are involved in the rupture of the lymph gland and the differentiation of lamellocytes. Nevertheless, the mechanisms by which the Toll pathway is activated and how it mediates its effect remain elusive.

While the general function for Toll/NF-κB signaling has been studied in a number of publications previousl. This study has some strengths that go beyond the initial correlations of the pathway activity with phenotypes and could therefore make an interesting advance for publication at *eLife*. Yet, for this to be so some open questions need to be addressed and the presentation/focus of the paper must be significantly improved. The manuscript therefore has potential, with appropriate additions and modifications but these need to be robustly addressed.

Essential revisions:

1) The Toll pathway mutants used in this study have different genetic backgrounds. To avoid any background effect (and taking in consideration that the Dif mutation has been subjected to reevaluation (Le Bourg 2011)), I would recommend the authors to use isogenic fly stocks. Importantly, isogenic fly stocks for *Dif, pll* and *dl* have already been produced by the Teixeira lab (Ferreira, 2014). In addition, it would be interesting to know if silencing Toll itself gave the same phenotype as silencing of the downstream components and SPE? Although it sounds trivial, this may be an important question to understand the upstream regulation of the pathway in the PSC.

2) What remains unclear is whether mutants or knockdowns of the Toll pathway components indeed block the response altogether, or rather delay the response? This should be addressable by examining the LG at later time points following wasp egg parasitism. In the same line, it would be interesting to know what the contribution of the Toll pathway in the disruption of the LG at the early pupal stage is.

3) Lamellocytes can originate from both sessile/circulating hemocytes and the lymph gland. The authors make the point that this study proves that the LG is a major source of lamellocytes, somewhat contrasting a previous publication by Markus et al. showing that the lamellocyte response can be largely mediated by the embryonic lineage of hemocytes in the larva. This statement is a little difficult to take for granted if the authors do not provide any data of embryonic lineage hemocytes in the larva, in parallel to the LG data. Compared to the LG differentiation, when do the authors find lamellocyte differentiation of the embryonic lineage (assuming that is occurs at some point)? Can the authors describe what might be the difference between their own experiments and the study by Markus et al.? For example, is there a difference in the nutritional status or temperature under which the experiments are being carried out? Clarifying these apparently contradicting points would be important to consolidate the findings from the different labs and thereby provide much needed insight and important guidelines for researchers in the field. This is especially important since many GAL4 line have broader expression pattern than expected. Thus, it could not be excluded that the phenotype observed (lack of encapsulation) is also linked to a defect in sessile/circulating hemocytes. Therefore, the authors should analyze lamellocyte differentiation in the hemolymph compartment (and more globally the state of the sessile/circulating hemocyte compartments). It would be interesting to decipher the respective function of Toll in the PSC, LG progenitors, circulating/sessile hemocytes and the fat body in the cellular response by using specific driver. Can lamellocyte differentiation be blocked by silencing Toll activity in circulating/sessile hemocytes or in the fat body? Such an experiment will add a lot in our understanding of the cellular response.

4) The authors use wasp production as readout for the effectiveness of the lamellocyte/encapsulation response. While presumably other immune cell types also play a role, this may nevertheless be a reasonable measure to quantify the response.

Since this seems a readout that is not used routinely in many papers, it would be helpful to emphasize the strategy and establish this assay formally in the manuscript. In later figures, this could then be used as a reference point. More precisely, is the first set of experiments meant to establish a strict relationship, sequentially linking lamellocyte differentiation and numbers in the lymph gland, to the rate of lymph gland breakdown and to encapsulation success? If so, this must be declared explicitly because it justifies why the rest of the experiments only measure the latter two. From Figure 1–Figure 2 onwards, lamellocytes are no longer quantified and there is an assumption that success of encapsulation and/or lymph gland breakdown correlates with the hematopoietic control, namely lamellocyte differentiation. Although this assumption might be correct it must be declared and sustained in a more explicit manner.

5) Mechanism of Toll activation: The authors should provide more information on the mechanism of Toll activation in the PSC. According to current model, Spz and SPE need to be cleaved by a serine protease. Is the activation of the Toll pathway dependent of Psh, ModSP, or PGRP-SA/GNBP1? All the mutants to test this hypothesis are available including Psh, ModSp double mutants.

6) The source of ROS could be identified as good RNAi lines exist for Duox or Nox. ROS could be visualized using a Kit. The catalase used is expressed intracellularly. Can they reproduce the phenotype by using an extra cellular catalase? How can an intracellular ROS burst lead to the activation of an extracellular serine protease?

7) It is unclear whether the role of Toll in the PSC is to promote the rupture of the LG or to stimulate the differentiation of lamellocyte, the latter inducing the rupture of the lymph gland. Can they separate the two processes? This point requires at least further discussion. What is the function of Toll pathway activation in progenitor cells?

8) Many figures lack statistical analysis.

Figure 1: and every other disruption quantification lacks statistical analysis. As convincing as they are, a simple χ^2^ on the raw numbers should provide unequivocal confirmation of the statements made. Idem for Figure 1: in panels L-Q there is no quantification of lamellocytes comparing the different conditions.

[Editors' note: further revisions were requested prior to acceptance, as described below.]

Thank you for resubmitting your work entitled "Reactive oxygen species-dependent Toll/NF-κB activation in the *Drosophila* hematopoietic niche confers resistance to wasp parasitism" for further consideration at *eLife*. Your revised article has been favorably evaluated by K VijayRaghavan (Senior editor), a Reviewing editor, and two of the three original referees. The decision is to request specific revisions to increase the robustness of your findings. Details are included below and these issues need to be addressed before we recommend acceptance.

Essential Revisions

1) Figure 1 (after N-Q), it is important to show an image or quantification showing lymph gland disruption in *Dif* and *Pell* mutant at 48h.

2) Psh and ModSp (Figure 5—figure supplement 1) the double mutant should be included. The role of Psh should be investigated in the context of LG disruption and not only in the context of wasp encapsulation. The phenotype could be explained by a role of Psh in the melanization reaction around the wasp and not LG disruption.

3) Do a deficiency in ROS or EGF-R block LG dispersal or just delay the process like a Toll mutant? The authors assume the effect is the same, which seems like a reasonable assumption given the relationship they have between the pathways. Nevertheless the paper will be reinforced if they can document this

4) Reading the revised version, the reviewers better realize that Toll mutants do not block lymph gland dispersal but rather delay it. This was not so clear in the original version since the authors focus at the 30h time point. We encourage the authors to highlight this point throughout the paper and discuss it in the Discussion.

Discussion should underline the following points:a) How the Lymph gland disrupt in Toll deficient mutant? It is important to underline that the processed is delayed but not block.b) How an intracellular burst of ROS can lead to the activation of extracellular serine protease cascade?

---

## [Author Response]

Essential revisions:1) The Toll pathway mutants used in this study have different genetic backgrounds. To avoid any background effect (and taking in consideration that the Dif mutation has been subjected to reevaluation (Le Bourg 2011)), I would recommend the authors to use isogenic fly stocks. Importantly, isogenic fly stocks for Dif, pll and dl have already been produced by the Teixeira lab (Ferreira, 2014). In addition, it would be interesting to know if silencing Toll itself gave the same phenotype as silencing of the downstream components and SPE? Although it sounds trivial, this may be an important question to understand the upstream regulation of the pathway in the PSC.

We have now studied lymph gland disruption and wasp egg hatching in the isogenic mutants for *Dif*, *pll, dl* as well as *Toll*, generated by Teixeira and col. This analysis confirmed our previous results for *dif*, *pll* and *dl* mutants which were generated in different genetic backgrounds, and further confirmed that *Toll* is required for wasp encapsulation as previously reported by Sorrentino et al., 2004. These additional data are reported in Figure 1—figure supplement 2, Figure 2—figure supplement 1 and in the Results section.

2) What remains unclear is whether mutants or knockdowns of the Toll pathway components indeed block the response altogether, or rather delay the response? This should be addressable by examining the LG at later time points following wasp egg parasitism. In the same line, it would be interesting to know what the contribution of the Toll pathway in the disruption of the LG at the early pupal stage is.

This is an important point. In all mutants and knock down conditions where we observed a delay in the response of the lymph gland to wasp egg laying, e.g., lamellocyte differentiation and disruption, the hematopoietic organ ultimately disrupts. This indicates that knocking down the Toll pathway does not block the lymph gland response but induces a delay in this response. These data are now reported in Figure 1—figure supplement 1 and in the Results section.

3) Lamellocytes can originate from both sessile/circulating hemocytes and the lymph gland. The authors make the point that this study proves that the LG is a major source of lamellocytes, somewhat contrasting a previous publication by Markus et al. showing that the lamellocyte response can be largely mediated by the embryonic lineage of hemocytes in the larva. This statement is a little difficult to take for granted if the authors do not provide any data of embryonic lineage hemocytes in the larva, in parallel to the LG data. Compared to the LG differentiation, when do the authors find lamellocyte differentiation of the embryonic lineage (assuming that is occurs at some point)?

We now analysed the percentage of lamellocytes (relative to total hemocytes) in the hemolymph of individual parasitized larvae prior to, and immediately after, lymph gland dispersal. For this analysis, we first determined for each genotype analyzed (the hml> control and *Dif1* mutant) the time window of lymph gland disruption and performed individual bleeding of larvae in this time window. Second, and this is crucial, we dissected each bled larva to check for 1) the presence of a wasp egg (confirming that it has been infected) and 2) the status of the lymph gland (intact or disrupted). The hml> control and *Dif1* mutant displayed different timings of lymph gland disruption (20H and 30H post parasitism, respectively). Only few circulating lamellocytes (less than 10% of all circulating hemocytes), most likely differentiating from embryonic sessile/circulating hemocytes, were observed before lymph gland disruption in both the control and mutant strains. The percentage of circulating lamellocytes increased sharply when lymph glands started disrupting to reach >50% of all circulating hemocytes after full disruption. We were thus able to conclude that lymph gland disruption contributes a major fraction of circulating lamellocytes post parasitism. These additional data are provided in Figure 2 and described in subsection “Lymph gland lamellocytes are required for successful wasp egg encapsulation”.

Can the authors describe what might be the difference between their own experiments and the study by Markus et al.? For example, is there a difference in the nutritional status or temperature under which the experiments are being carried out? Clarifying these apparently contradicting points would be important to consolidate the findings from the different labs and thereby provide much needed insight and important guidelines for researchers in the field.

Markus et al., 2009 have reported that lymph gland anterior lobes remain intact until 48H post parasitism in a “control” Hemese-GAL4, UAS-GFP.nls strain. They performed wasp infection overnight, at 18°C. In our hands also, when parasitized larvae are grown at 18°C, lymph gland disruption is delayed compared to larvae grown at 27°C or 29°C, somewhat expectedly since larval development altogether is slower. In our assays, wasp infection was allowed for only 1H at 22°C and then the larvae were grown either at 27°C or at 29°C. We observed that control lymph glands disperse from 20H post parasitism onwards and at the latest 30H post parasitism, depending on the genetic background. More importantly, Markus et al. did not perform, as we did, any wasp egg hatching assay in their conditions. Such a test is crucial to validate that the kinetics of lamellocyte differentiation allow successful wasp egg encapsulation.

In the revised version of the manuscript, we have added data relative to the delay of the lymph gland response when larvae are grown at 18°C, in subsection “Lymph gland lamellocytes are required for successful wasp egg encapsulation” and Figure 2—figure supplement 2. In addition, we have discussed in paragraph three of the Discussion our data relative to those previously published by Markus et al.

This is especially important since many GAL4 line have broader expression pattern than expected. Thus, it could not be excluded that the phenotype observed (lack of encapsulation) is also linked to a defect in sessile/circulating hemocytes.

We impaired lymph gland lamellocyte differentiation and lymph gland disruption using the two drivers *domeMESO*> (*domeMESO*> *lat RNAi*) and dome> (*dome>catalase*) which are expressed in lymph gland progenitors and not in sessile/circulating hemocytes (data not shown). In these two contexts, an enhancement of wasp egg hatching was observed, supporting the conclusion that successful host defense against wasp parasitism is linked to the lymph gland response. These data are provided in Figure 2 and Figure 6.

Therefore, the authors should analyze lamellocyte differentiation in the hemolymph compartment (and more globally the state of the sessile/circulating hemocyte compartments).

As mentioned above in the response to point 3, we quantified the% of circulating lamellocytes in individual parasitized larvae and checked in parallel the status of lymph glands (intact or disrupted) (Figure 2). Of note, a recent paper (Anderl et al., 2016) from the lab of Dan Hultmark, (a co-author of Markus et al., 2009), established that no major change of sessile hemocyte organisation occurs in response to parasitism.

It would be interesting to decipher the respective function of Toll in the PSC, LG progenitors, circulating/sessile hemocytes and the fat body in the cellular response by using specific driver. Can lamellocyte differentiation be blocked by silencing Toll activity in circulating/sessile hemocytes or in the fat body? Such an experiment will add a lot in our understanding of the cellular response.

The function of the Toll pathway in the PSC and lymph gland progenitors is documented in Figure 4 and Figure 4—figure supplement 1, respectively. Schmid et al., 2014 investigated the requirement of the Toll pathway in the fat body and sessile hemocytes for wasp egg encapsulation. To inactivate the Toll pathway, they expressed *Myd88* RNAi and *pelle* RNAi in the fat body and/or sessile/circulating hemocytes using the FB-Gal4 and Hml-Gal4 drivers, respectively. They measured the% of killed wasp parasites and wasp egg encapsulation (Figure 5 in Schmid et al., 2014). They concluded that the Toll pathway is not required in these two tissues for the response to wasp parasitism. We further addressed this question by analysing both lymph gland lamellocyte differentiation and lymph gland dispersal in parasitized larvae in which the Toll pathway is knocked down in either hemocytes (*Hml-Gal4>Myd88 RNAi*) or fat body (*FB-Gal4> Myd88 RNAi*). No effect was observed, indicating that the Toll pathway is not required in hemocytes or fat body for proper lymph gland response to parasitism. Since our data confirm Schmid et al. results, we did not include them in a supplementary figure but reported on them in the Discussion as data not shown.

4) The authors use wasp production as readout for the effectiveness of the lamellocyte/encapsulation response. While presumably other immune cell types also play a role, this may nevertheless be a reasonable measure to quantify the response.Since this seems a readout that is not used routinely in many papers, it would be helpful to emphasize the strategy and establish this assay formally in the manuscript.

Wasp larvae production has been used as a functional test to measure wasp egg encapsulation in several other recent articles: Vanha-aho et al., 2015; Schmid et al., 2014; Mortimer et al., 2013,. These references are now added in the Results section to emphasize that this criterion is currently used by many labs.

In later figures, this could then be used as a reference point. More precisely, is the first set of experiments meant to establish a strict relationship, sequentially linking lamellocyte differentiation and numbers in the lymph gland, to the rate of lymph gland breakdown and to encapsulation success? If so, this must be declared explicitly because it justifies why the rest of the experiments only measure the latter two. From Figure 1–Figure 2 onwards, lamellocytes are no longer quantified and there is an assumption that success of encapsulation and/or lymph gland breakdown correlates with the hematopoietic control, namely lamellocyte differentiation. Although this assumption might be correct it must be declared and sustained in a more explicit manner.

This is a very important point, indeed. To challenge the assumption that lymph gland breakdown and encapsulation success correlate with lamellocyte differentiation in the lymph gland, we looked at lamellocyte differentiation in several genetic contexts where lymph gland dispersal is either impaired such as *col>Myd88 RNAi* or *col>catalase*, or similar to control such as *dome>Myd88 RNAi*. Our data allow us to conclude that encapsulation success and/or lymph gland breakdown correlate with lymph gland lamellocyte differentiation. This result is now better documented in Figure 4, Figure 4—figure supplement 2 and Figure 6—figure supplement 2, and stated more explicitly in the Results section.

5) Mechanism of Toll activation: The authors should provide more information on the mechanism of Toll activation in the PSC. According to current model, Spz and SPE need to be cleaved by a serine protease. Is the activation of the Toll pathway dependent of Psh, ModSP, or PGRP-SA/GNBP1? All the mutants to test this hypothesis are available including Psh, ModSp double mutants.

Following the suggestions of the reviewer, we performed wasp egg encapsulation assays in *psh^Anderl et al., 2016^*, *modSP^Anderl et al., 2016^* and *psh^Anderl et al., 2016^;modSP^Anderl et al., 2016^* double mutants. Whereas no defect was observed in *modSP^Anderl et al., 2016^*mutants, *psh^Anderl et al., 2016^* and *psh^Anderl et al., 2016^;modSP^Anderl et al., 2016^*mutants failed to encapsulate wasp eggs. These new data on *psh^Anderl et al., 2016^* and *modSP^Anderl et al., 2016^*mutants are shown in Figure 5—figure supplement 1 and mentioned in the Results section.

6) The source of ROS could be identified as good RNAi lines exist for Duox or Nox. ROS could be visualized using a Kit. The catalase used is expressed intracellularly. Can they reproduce the phenotype by using an extra cellular catalase? How can an intracellular ROS burst lead to the activation of an extracellular serine protease?

PSC ROS levels were analysed 6H post parasitism, in larvae where Duox or Nox were specifically downregulated in PSC cells (col>Duox RNAi and col>Nox RNAi, respectively) or where extracellular catalase was overexpressed (col>UAS-IRC). In none of these contexts did we observe a difference in ROS levels in PSC cells, indicating that they are generated by PSC cells. These new data are reported in Figure 6—figure supplement 1 and in the Results section. Deciphering how ROS activates Toll/NF-κB in PSC cells requires further considerable investigations and is out of scope in the present article.

7) It is unclear whether the role of Toll in the PSC is to promote the rupture of the LG or to stimulate the differentiation of lamellocyte, the latter inducing the rupture of the lymph gland. Can they separate the two processes? This point requires at least further discussion. What is the function of Toll pathway activation in progenitor cells?

We showed that Dif mutants exhibit a delay of lymph gland lamellocyte differentiation correlating with a delay in lymph gland dispersal (Figure 1). We have now analysed lymph gland lamellocyte differentiation when the Toll pathway is specifically inactivated in the PSC and, as reported in Dif mutants (see response to point 4), we found a delay in lymph gland lamellocyte differentiation associated with the defect of lymph gland disruption. Together, these data strongly suggest that Toll activation in the PSC promotes rapid differentiation of lamellocytes in the lymph gland that in turn leads to lymph gland dispersal.

8) Many figures lack statistical analysis.Figure 1: and every other disruption quantification lacks statistical analysis. As convincing as they are, a simple χ^[2]^ on the raw numbers should provide unequivocal confirmation of the statements made. Idem for Figure 1: in panels L-Q there is no quantification of lamellocytes comparing the different conditions.

As recommended by the reviewer, we performed χ^2[2]^ tests on all lymph gland disruption and wasp egg hatching assays as well as in Figure 1; panels L-Q. These data were added in the revised version.

[Editors' note: further revisions were requested prior to acceptance, as described below.]

Essential Revisions1) Figure 1 (after N-Q), it is important to show an image or quantification showing lymph gland disruption in Dif and Pell mutant at 48h.

We have added images of lymph gland disruption in *Dif* and *pelle* mutants 48H post parasitism (Figure 1, panels H, I).

2) Psh and ModSp (Figure 5—figure supplement 1) the double mutant should be included. The role of Psh should be investigated in the context of LG disruption and not only in the context of wasp encapsulation. The phenotype could be explained by a role of Psh in the melanization reaction around the wasp and not LG disruption.

We have added the quantification of wasp egg hatching in the double *psh^Anderl et al., 2016^; modSP^Anderl et al., 2016^* mutant. We have also examined lymph gland disruption in the *Psh^Anderl et al., 2016^*mutant and found a delay compared to control larvae. These data are added in Figure 5—figure supplement 1.

3) Do a deficiency in ROS or EGF-R block LG dispersal or just delay the process like a Toll mutant? The authors assume the effect is the same, which seems like a reasonable assumption given the relationship they have between the pathways. Nevertheless the paper will be reinforced if they can document this

In both mutant conditions (ROS suppression in PSC cells or EGFR downregulation in progenitors), 100% of lymph glands are disrupted 48H post parasitism, indicating a delay compared to control. These data are now reported in an additional figure (Figure 6—figure supplement 3).

4) Reading the revised version, the reviewers better realize that Toll mutants do not block lymph gland dispersal but rather delay it. This was not so clear in the original version since the authors focus at the 30h time point. We encourage the authors to highlight this point throughout the paper and discuss it in the Discussion.

We have now modified the text throughout the paper to emphasize that in all mutants and knock down conditions analysed, we observed a delay in lymph gland disruption compared to control.

Discussion should underline the following points:a) How the Lymph gland disrupt in Toll deficient mutant? It is important to underline that the processed is delayed but not block.

We discuss in the revised version of the article how Toll/NF-κB signalling could promote lymph gland dispersion, in light of a recent publication proposing that this pathway might regulate the permeability barrier of PSC cells (Khadilkar et al., 2017) (see Discussion, fourth paragraph).

b) How an intracellular burst of ROS can lead to the activation of extracellular serine protease cascade?

We have found that Toll/NF-κB activation in the lymph gland requires SPE production from PSC cells (Figure 5). Thus, an intracellular burst of ROS in PSC cells could activate SPE processing/secretion from these cells and trigger pathway activation in the lymph gland. Alternatively, the extracellular protease cascade leading to Spätzle cleavage could be independent of the oxidative stress occurring in PSC cells, and ROS might regulate an intracellular step of Toll/NF- κB activation in the PSC, such as modifying iκB kinase complex (IKK) activity in the cytoplasm or NF-κB DNA binding capacity in the nucleus, as reported in mammals. These two possibilities are now proposed (see Discussion section).